# Deep Learning for Gravitational-Wave Data Analysis: A Resampling White-Box Approach

**DOI:** 10.3390/s21093174

**Published:** 2021-05-03

**Authors:** Manuel D. Morales, Javier M. Antelis, Claudia Moreno, Alexander I. Nesterov

**Affiliations:** 1Departamento de Física, Centro Universitario de Ciencias Exactas e Ingenierías, Universidad de Guadalajara, Av. Revolución 1500, Guadalajara 44430, Mexico; claudia.moreno@cucei.udg.mx (C.M.); nesterov@cencar.udg.mx (A.I.N.); 2Tecnologico de Monterrey, School of Engineering and Science, Monterrey, NL 64849, Mexico; mauricio.antelis@tec.mx

**Keywords:** gravitational waves, Deep Learning, convolutional neural networks, binary black holes, LIGO detectors, probabilistic binary classification, resampling regime, white-box testings, uncertainty, 04.30.-w, 07.05.Kf, 07.05.Mh

## Abstract

In this work, we apply Convolutional Neural Networks (CNNs) to detect gravitational wave (GW) signals of compact binary coalescences, using single-interferometer data from real LIGO detectors. Here, we adopted a resampling white-box approach to advance towards a statistical understanding of uncertainties intrinsic to CNNs in GW data analysis. We used Morlet wavelets to convert strain time series to time-frequency images. Moreover, we only worked with data of non-Gaussian noise and hardware injections, removing freedom to set signal-to-noise ratio (SNR) values in GW templates by hand, in order to reproduce more realistic experimental conditions. After hyperparameter adjustments, we found that resampling through repeated *k*-fold cross-validation smooths the stochasticity of mini-batch stochastic gradient descent present in accuracy perturbations by a factor of 3.6. CNNs are quite precise to detect noise, 0.952 for H1 data and 0.932 for L1 data; but, not sensitive enough to recall GW signals, 0.858 for H1 data and 0.768 for L1 data—although recall values are dependent on expected SNR. Our predictions are transparently understood by exploring tthe distribution of probabilistic scores outputted by the softmax layer, and they are strengthened by a receiving operating characteristic analysis and a paired-sample t-test to compare with a random classifier.

## 1. Introduction

Beginning with the first direct detection of a gravitational wave (GW) by LIGO and Virgo collaborations [1], the observation of GW events that are emitted by compact binary coalescences (CBCs) have made GW astronomy, to some extent, a routine practice. 11 of these GW events were observed during O1 and O2 scientific runs, which are collected in the catalogue GWTC-1 [2] of CBCs and, more recently, 39 events have been reported in the new catalogue GWTC-2 [3] of CBCs, which are detections during the first half of the third scientific run, namely O3a. This last run have been recorded with the network of LIGO Livingston (L1) and Hanford (H1) twin observatories and Virgo (V1) observatory; an, recently, KAGRA observatory joined the network of GW detectors [4].

The sensitivity of GW detectors has been remarkably increased these last years and, in this improvement process, GW data analysis had a crucial role quantifying and minimizing, as much as possible, the effects of non-Gaussian noise—mainly short-time transients, called “glitches” [5], for which, in several cases, such as those involving blip transients [6], there is still no full explanations about their physical causes. Monitoring signal-to-noise ratio (SNR) of all detections is an essential procedure here, indeed being the heart of the standard algorithm that is used for the detection and characterization of GWs in LIGO and Virgo, namely Matched Filter (MF) [7,8,9]. This technique, which is the GW detection stage of current CBC search pipelines of LIGO, is based on the assumption that signals to be detect are embedded in additive Gaussian noise and it consists in a one-by-one correlation between data from detectors (after a whitening process) with each one GW template of a bank of GW templates. Subsequently, those matchs that minimize SNR [10] are selected. Standard pipelines that include MF and are used in LIGO and Virgo analyses are PyCBC [11,12], GstLAL [13], and MBTA [14].

MF has shown be a powerful tool having a crucial role in all confirmed detections of GWs that are emitted by CBCs. However, there are significant reasons for exploring alternative detection strategies. To begin, noise outputted by interferometric detectors is non-Gaussian, conflicting with assumption of Gaussian noise in MF. Generally, and also included in LIGO search pipelines, supplementary tools are used for facing this issue as whitening techniques [15]. However, as glitches remain, consistence statistical tests [16] and coincidence procedures as cross-correlation [17,18] are applied for a network of detectors, raising the still open problem of how to systematically deal with single-interferometer data. In sum, the assumption about Gaussian noise of MF raises the issue about how to detect glitches in single-interferometer data.

Elsewhere, one would want to perform more general GW searchings. The natural next step is to include precessing spins [19], orbital eccentricity [20], and neutron star tidal deformability [21], among other aspects. The more general a GW searching the much wider the parameter space and MF becomes computationally prohibitive.

Actually, if we are not interested in replacing MF, still it is pertinent to have alternative algorithms just for independent verifications and/or increasing the confidence level of GW detections.

In this context, Machine Learning (ML) [22] and its successor, Deep Learning (DL) [23], emerge as promising alternatives or, at least, complementary tools to MF. These techniques assume nothing regarding background noise, which is clearly advantageous if we want to work with single-interferometer data; we can identify non-Gaussian noise without needing a network of detectors. In addition, they have shown be remarkably useful in analyzing enormous mounts of data through sequential or online learning processes, which could be a significant improvement for more general GW searchings in real time. Additionally, if we do not have deterministic templates to predict GW signals as those emmited by Core-collapse supernovae (CCSNe) or anticipate noise artifacts as non-Gaussian glitches, unsupervised ML and DL algorithms (which do not need prior labeled training samples) could be interesting for future explorations.

The implementation of ML algorithms in GW data analysis has no more than a few years. Biswas et al. [24] contributed with a pioneering work, where Artificial Neural Networks (ANNs), Support Vector Machines, and Random Forest algorithms were used to detect glitches in data from H1 and L1 detectors, recorded during S4 and S6 runs. Competitive performance results, back then, were obtained. Later, works focused on several problems were published, with better results. For instance: ANN for the detection of GWs that are associated with short gamma-ray bursts [25], Dictionary Learning for denoising [26], Difference Boosting Neural Network and Hierarchical Clustering for detection of glitches [27], ML and citizen science for glitches classification [28], and background reduction for CCSNe searching with single-interferometer data [29], among others.

Applications of DL, in particular Convolutional Neural Network (CNN) algorithms, are even more recent in GW data analysis. Gabbard et al. [30], George and Huerta [31], provided first works, in which CNNs were implemented to detect simulated GW signals from Binary black holes (BBHs) that were embedded in Gaussian noise. They claimed that performance of CNNs was similar and much better than MF, respectively. Later, George and Huerta extended their work by embedding simulated GWs and analytical glitches in real non-Gaussian noise data of LIGO detectors, with similar results [32]. Thereafter, CNN algorithms has been applied for several instrumental and physical problems, showing more improvements. For instance, the detection of glitches [33], trigger generation for locating coalescente time of GWs emitted by BBHs [34], detection of GWs from BBHs [35] and Binary neutron star (BNS) systems [36], detection of GWs emitted by CCSNe and using both phenomenological [37] and numerical [38] waveforms, and the detection of continuous GWs from isolated neutron stars [39], among others.

From a practical point of view, previous works of CNN algorithms that were applied to GW detection and characterization reach competitive performance results according to standard metrics, such as accuracy, loss functions, false alarm rates, etc., showing feasibility of DL in GW data analysis in the first place. However, from a formal statistical point of view, we warn that, beyond just applications, we are in need of deeper explorations that seriously take the inevitable uncertainty that is involved in DL algorithms into account before putting them as real alternatives to standard pipelines in LIGO and Virgo.

For this research, our general goal is to draw on CNN architectures to make a standard GW detection. In particular, beginning from a training set containing single-interferometer strain data, and then transforming it into time-frequency images with a Morlet wavelet transform, the aim is distinguish those samples that are only non-Gaussian noise from those samples that contain a GW embedded in non-Gaussian noise—in ML language, this is just a binary classification. Moreover, as a novel contribution, here we will take two ingredients into consideration to advance towards a statistically informed understanding of involved uncertainties, namely: resampling, and a white-box approach. Resampling consists of repeated experiments for training and testing CNNs, and the white-box approach involves a clear mathematical understanding of how these CNNs internally work. See Section 2.4 for more details.

Our choice of Morlet as a mother wavelet is because we want to capture all time-frequency information of each sample at once rather performing multiresolution analyses for detecting closely space features, given that our CNN algorithms, precisely, will be in charge of performing feature extraction for our image classification. Subsequently, this setup removes, beforehand, the need of working with compact support wavelets for multiresolution analyses such as Haar, Symlet, and Daubechies, among others [40]. Our Morlet wavelet is also normalized to conserve energy and become symmetric.

In this work, we are not interested in reaching higher performance metric values than those reported in previous works, neither testing new CNN architectures, nor using latest real LIGO data and/or latest simulated templates, but rather facing the question about how to clever deal with uncertainties of CNNs, which is an unescapable requesite if we claim that DL techniques are real alternatives to current pipelines. Moreover, as a secondary goal, we want to show that CNN algorithms for GW detection, even when considering repeated experiments, can be easily run in a single local CPU and reach good performance results. The key motivation for this is to make reproducibility easier for the scientific community. We also released all of the code used in this work [41].

With regard to data, we decided to use recordings from S6 LIGO run, separately from H1 and L1 detectors, considering GW signals of CBCs that were only generated by hardware injections. This choice is ultimately motivated by the fact that for reaching stronger conclusions, CNN algorithms should be tested in conditions as adverse as possible. To work solely with hardware injections means to eliminate freedom to set SNR values by hand and, therefore, to remove a choice that can influence performance of CNN algorithms. Subsequently, we draw on S6 data because they have a greater amount of hardware injections than more recent data. See Section 2.2 for more details.

## 2. Methods and Materials

### 2.1. Problem Statement

As starting point of our problem, we have a i−th slice of raw strain data recorded at one of the LIGO detectors. In mathematical notation, this slice of data is expressed by a column vector of times series in *N* dimensions,
(1)$$\begin{array}{ccc} s_{raw}{}^{i}\left(t\right) & = & {\left[s\left(t_{0}{}^{i}\right),s\left(t_{1}{}^{i}\right),...,s\left(t_{N-1}{}^{i}\right)\right]}^{T}\;, \end{array}$$
where the sampling time is $t_{s}=t_{j}{}^{i}-t_{j-1}{}^{i}$ with $j=1,2,...,N_{slice}-1$, the sampling frequency is $f_{s}=1/t_{s}$, and, of course, the time length of the slice is $T_{slice}=N_{slice}/f_{s}$ (in seconds) with $N_{slice}$ points of data. The next step is to theoretically model the above slice of data; therefore, we introduce the following expression:(2)$$\begin{array}{c} s_{raw}{}^{i}\left(t\right)=\left\{\begin{array}{cc} n^{i}\left(t\right) & \mathrm{if}\;\mathrm{there}\;\mathrm{is}\;\mathrm{not}\;a\;\mathrm{GW}\;,\\ n^{i}\left(t\right)+h^{i}\left(t\right) & \mathrm{if}\;\mathrm{there}\;\mathrm{is}\;a\;\mathrm{GW}\;, \end{array}\right. \end{array}$$
where $n^{i}\left(t\right)$ is the non-Gaussian noise from the detector and $h^{i}\left(t\right)$ is the observed strain, which is a function of the antenna response of the interferometer when a GW, with unknown duration, arrival time, and waveform, is detected. For this research, the problem that we address is to decide whether a segment of strain data contains only noise, or it contains noise plus a unknown GW signal. Then, in practice, we will implement CNN algorithms that performs a binary classification, inputting strain samples of time lengh $T_{win}<T_{slice}$ and deciding, for each sample, if it does not contain a GW signal (i.e., the sample ∈ class 1), or it contains a GW signal (i.e., the sample ∈ class 2).

### 2.2. Dataset Description

For this study, we use real data that were provided by LIGO detectors, which are freely available on the LIGO-Virgo Gravitational Wave Open Science Center (GWOSC), https://www.gw-openscience.org (accessed on 1 April 2019) [42]. Because of reasons that are presented at the end of Section 1, we decided to use data of sixth science run (S6), recorded from 7 July 2009 to 20 October 2010. During this run, the detectors achieved a sensitivity given by a power spectral denstity around $10^{-22}$, with uncertainties up to ±15% in amplitude [43]. Each downloaded strain data slice has a time length of $T_{slice}=4096$ s and a sampling frequency of $f_{s}=4096$ Hz.

S6 contains hardware injections that were already added to the noise strain data. These injections are generated by actuating the test masses (mirrors) and, therefore, changing the physical length of the arms of the detectors. This procedure simulates the effects of GWs, and is used for experimental tests and calibration of the detectors [44]. For this research, we solely work with hardware injections of GWs emitted by CBCs: 724 injections in data from H1 detector and 656 injections in data from L1 detector. For each injection, we know the coalescence (or merger) time $t_{c}$ in Global Positioning System (GPS), masses $m_{1}$ and $m_{2}$ in solar mass units $M_{⊙}$, the distance *D* to the source in Mpc, and the expected and the recovered signal-to-noise ratio, namely $SNR_{exp}$, $SNR_{rec}$, respectively. All of this information is provided by LIGO on the aforementioned website.

Because S6 has a much greater mount of hardware injections of CBC GW signals than in later public data, this is the best option for our study. Besides, if we reach good results with S6 data, we are providing highly convincing evidence for GW detection by using CNNs because the sensitivity of S6 is less than in later public data and, even, we could expect that working with later public data our algorithms will work much better.

Take in mind that our approach allows for advancing towards more realistic experimental conditions regarding the evaluation of the CNNs. We are showing, beyond any doubt, that we do not have control on distribution of recorded real signals and, therefore, we do not have any ability to influence the performance results. It is known that the SNR values of recorded GW events and their frequency of occurrence are given as (not handleable) facts. Along a LIGO run data, both in hardware injections set and real observing signals set, each event consecutively occurred in real time once and never again. Therefore, if our CNN algorithms are able to deal with this starting adverse evaluation constraint, then we are advancing towards the more general purpose of analyzing arbitrarily distributed data without drawing on ad hoc choices that could generate excessively optimistic performance results.

The above advantage does not lie on the set of parameters that characterize hardware injections, which is to say, data populations. In general, hardware injections has been used to perform end-to-end validation checks [45], and they are experimentally expensive to perform as repeatedly as software injections to generate diverse distributions of GW events. The point rather is that our approach lies on the fact that we cannot vary these populations by hand, that is to say, on the constraint to generate this population, not in the population itself. This advantage cannot achieved with software injections, which still are handleable and, therefore, the evaluation results can be influenced. For instance, in random software injections, we could vary injection ratio to influence the trade-off between variance vs. bias of stochastic predictions and, therefore, to improve performance of a CNN algorithm either by artificially balancing dataset regarding target values and/or costs of different classification errors.

It should be stressed that problem of bias and fairness (with regards to ad hoc choices in datasets and algorithms) is an open and highly debate issue in the ML community [46], even with deep ethical implications if we apply ML/DL to social ambits. Of course, in GW data analysis, we do not have this kind of implications, but this does not remove the fact that, at a pure statistical level, bias and fairness are still present. If we are committed to a deep multidisciplinary understanding of ML, it is needed to face this challenge by drawing on evaluations as transparent as possible.

In any case, even if the above advantage is not considered, our approach still is scientifically relevant, because it lays the groundwork for future calibration procedures with CNN detection algorithms inputting hardware injections.

Figure 1 shows the distributions of injections present in S6 data from H1 and L1 detectors, detaling information about total mass $M=m_{1}+m_{2}$ of each source and distance *D* to it. These injections simulate stellar BBHs that are located at the scale of our galaxy supercluster, covering distances from 1 Mpc to 100 Mpc and masses from $2.8M_{⊙}$ to $35M_{⊙}$. In both panels, we have a high occurrency of events of masses from $1M_{⊙}$ to $12M_{⊙}$ at distances from 1 Mpc to 50 Mpc aprox. Even, in each of these regions there is a subregion where the highest concentration of events appears, namely for masses from $1M_{⊙}$ to $7M_{⊙}$ and distances from 1 Mpc to 26 Mpc aprox., corresponding to the scale of nearby galaxy clusters. Subsequently, in the region $M>12M_{⊙}$ and D<50 Mpc, we observe a clear decrease in the frequency of occurence of BBHs, which is even more pronounced for the farest events at distances D>50 Mpc for all masses. In any case, by solely exploring marginal histograms for distances, the clearest trend that we notice in data is that, as *D* increases from 4 Mpc, GW events are less frequent.

We downloaded 722 slices of strain data from the H1 detector and 652 slices of strain data from L1 detector, to be pre-process as explained in Section 2.3. Besides, we only consider hardware injection for which we know their parameters, because we will characterize detected GW events depending on their given expected SNR values, as will shown in results of Section 3.3.

### 2.3. Data Pre-Processing

With LIGO raw strain data segments at hand, the next methodological step is data pre-processing. This has three stages, namely data cleaning, the construction of strain samples, and application of wavelet transform.

#### 2.3.1. Data Cleaning

Data cleaning or data conditioning is a standard stage for reducing the noise and flattening the power spectral density (PSD). It consists in three steps. Firstly, for each i−th slice of raw strain data that we introduced in Equation (Equation 1), segments around coalescence times $t_{c}$ are extracted via blackman window of time length 128 s. If a slice does not have an injection, then it is not possible to extract a segment of 128 s and, therefore, this slice does not go beyond this step. Moreover, of all segments containing injections, we discard those with NaN (not a number) entries. This rejection is valid, because NaN entries span time ranges from 2 s to 50 s, being greater than time resolutions $T_{win}$ that we use for building samples to be inputted by the CNNs (see Section 2.3.2), namely from 0.25 s to 2.00 s (see Section 3.2). Here, feature engineering techniques as imputation or interpolation are not suitable. At this point, we decrease the mount of our data segments from 722 to 501 for H1 detector, and from 652 to 402 for L1 detector.

Secondly, a whitening is applied to each l−th of these segments as:(3)$$s_{white}{}^{l}\left(t\right)=\sum_{k=1}^{N}\frac{{\tilde{s}}_{raw}{}^{l}\left(f_{k}\right)}{\sqrt{S_{raw}{}^{l}\left(f_{k}\right)}}e^{i2\pi tk/N}\;,$$
where $\tilde{s}{}_{raw}^{l}\left(f_{k}\right)$ is the N−point Discrete Fourier Transform (DFT) of the 128 s segment of raw strain data $s_{raw}{}^{l}\left(t\right)$, $S_{raw}{}^{l}\left(f_{k}\right)$ the N− point two-sided PSD of the raw data, *N* is the number of points of data in the 128 s raw strain segment, and *i* the imaginary unit. In theory, PSD is defined as the Fourier transform of the raw data autocorrelation. Subsequently, we implemented the Welch’s estimate [47], where we compute the PSD between 0 Hz and 2048 Hz at a resolution of 1/128 Hz applying Hanning-windowed epochs of time length 128 s with an overlap of 64 s. At the end, the goal of whitening procedure is to approximate strain data to a Gaussian stochastic process that is defined by the following autocorrelation:(4)$$R\left(\tau \right)=\langle s_{white}\left(t+\tau \right)s_{white}\left(\tau \right)\rangle =\sigma^{2}\delta \left[\tau \right]\;,$$
with σ denoting the variance and δ[τ] the discrete time unit impulse function. (Explicitly: the discrete time unit impulse function is defined by δ[τ]=1 for τ=0, and δ[τ]=0 for τ≠0). Finally, a Butterworth band-pass filter from 20 Haz to 1000 Hz is applied to the already whitened segment. This filtering removes extreme frequencies that are out of our region of interest and discards 16 s on the borders of the segment to avoid spurious effects produced by the whitening, which results in a new segment $s_{white}+s_{bpf}=s_{clean}$ of time length $T_{clean}=96$ s. Edge effects appear because, in the whitening, DFT assumes that each finite slice $s_{raw}{}^{i}\left(t\right)$ is repeated in the form of consecutive bins, to have a fully periodic and time infinite data. This inescapable assumption leads to a spectral leakage by generating new artificial frequencies due to the inter-bin discontinuities.

Figure 2 shows how our strain data segments look after applying the whitening and the band-pass filtering, both in the time domain and amplitude spectral density (ASD), which is computed as $\sqrt{PSD}$. From the time domain plots, it can be seen that after the cleaning, amplitude of the strain data is reduced 5 orders of magnitude, from $10^{-16}$ to $10^{-21}$, and edge effects after whitening are clearly shown in the middle plot. Before applying cleaning, ASD shows the known noise profile that describes the sensitivity of LIGO detectors, which is the sum of contributions of all noise sources [48].

#### 2.3.2. Strain Samples

The building of strain samples that will be inputted by our CNNs is the next stage of the pre-processing. This is schematically depicted in Figure 3. This procedure has two steps. First, a l−th cleaned strain data segment, denoted as $s_{clean}{}^{l}\left(t\right)$, is splited in overlapped windows of duration $T_{win}$, identifying if it has or does not have an injection—time length $T_{win}$ is of the order of the injected waveform duration, leastwise. Here classes are assigned: if a window of data contains an injection, then class 2 (C2) is assigned; on the other hand, if that window does not contain an injection, class 1 (C1) is assigned. This class assignment is applied to all windows of data. Next, from the set of all tagged windows, and discarding beforehand those C1 windows with SNR<10, we select four consecutive ones of C1 and four consecutive ones of C2. This procedure is applied to each segment of clean data $s_{clean}{}^{l}\left(t\right)$ and, for avoid confusion with notation, we depict a k−th windowed strain sample as:(5)$$\begin{array}{ccc} s_{win}{}^{k}\left(t\right) & = & {\left[s\left(t_{0}{}^{k}\right),s\left(t_{1}{}^{k}\right),...,s\left(t_{N_{win}-1}{}^{k}\right)\right]}^{T}\;, \end{array}$$
where $N_{win}=f_{s}\;T_{win}$ according to the time length of the samples, with $f_{s}$ the sampling frequency. In practice, as we initially have 501 segments from H1 and 402 segments from L1, then 501×8=4008 and 402×8=3360 strain samples, respectively, are generated. Consequently, index *k* appearing in Equation (Equation 5) takes values from 0 to 4007 for H1 detector data, and values from 0 to 3359 for L1 detector data. Besides, $T_{win}$ is a resolution measure and, as it will seen later, we run our code with several values of $T_{win}$ in order to identify which of them are optimal with respect to the performances of CNNs.

#### 2.3.3. Wavelet Transform

Some works in GW data analysis have used raw strain time series directly as input to deep convolutional neural networks (e.g., Refs. [31,34]); however, we will not follow this approach. We rather decided to apply CNNs for what was designed in its origins [49], and for what have dramatically improved last decades [50], namely image recognition. Then, we need a method to transform our strain vectors to image matrices, i.e., grid of pixels. For this research, we decided to use the wavelet transform (WT), which, in signal processing, is a known approach for working in the time-frequency representation [51]. One of the great advantages of the WT is that, by using a localized wavelet as kernel (also called “mother wavelet”), it allows for visualizing tiny changes of the frequency structure in time and, therefore, improve the search of GW candidates that arise as non-stationary short-duration transients in addition to the detector noise.

In general, there are several wavelets that can be used as kernel. Here, we decide to work with the complex Morlet wavelet [52], which, in its discrete version, has the following form:(6)$$\psi \left(t_{n},f_{j}\right)=\frac{1}{\sqrt{\sigma^{t}{}_{j}\sqrt{\pi }}}\exp\left[\frac{-t_{n}{}^{2}}{2\left(\sigma^{t}{}_{j}\right){}^{2}}\right]\exp\left(2i\pi f_{j}t_{n}\right)\;,$$
having a Gaussian form along time and along frequency, with standard deviations $\sigma^{t}$ and $\sigma^{f}$, respectively. Moreover, these standard deviations are not independent of each other, because they are related by $\sigma^{t}{}_{j}=1/\left(2\pi \sigma^{f}{}_{j}\right)$ and $\sigma^{f}{}_{j}=f_{j}/\delta w$, where δw is the width of the wavelet and $f_{j}$ its center in the frequency domain.

It is important to clarify that Morlet wavelet has a Gaussian shape, then it does not have a compact support. For this reason, the mesh in which we defined Equation (Equation 6), i.e., the set of discrete values for time $t_{n}$ and frequency $f_{j}$, is infinite by definition. Further, resolutions $\Delta t=t_{n}-t_{n-1}$ and $\Delta f=f_{j}-f_{j-1}$ are solely constrained by our system resources and/or to what extent we want to economize these resources.

Subsequently, to perform the WT of the strain sample $s_{win}{}^{k}\left(t\right)$ (defined by Equation (Equation 5)) with respect the kernel wavelet $\psi (t_{n},f_{j})$ (defined by Equation (Equation 6)), we just need to compute the following convolution operation:(7)$$\begin{array}{ccc} Ws^{k}\left[t_{n},f_{j}\right] & = & \sum_{m=0}^{N_{win}-1}s^{k}\left(t_{m}\right)\psi^{*}\left(t_{m-n},f_{j}\right)\;, \end{array}$$
where $s^{k}\left(t_{m}\right)$ is the m−th element of the column vector $s_{win}{}^{k}\left(t\right)$. Besides, $n=0,1,...,N_{time}$ and $j=0,1,...,N_{freq}$, where $N_{time}$ and $N_{freq}$ define the size of each k−th image generated by the WT transform, being $Ws^{k}\left[t_{n},f_{j}\right]$ just the (n,j) element or pixel of the each generated image. In practice, we set our WT, such that it outputs images with dimensions $N_{time}=4096$ and $N_{freq}=47$ pixels.

In general, the grid of pixels that are defined by all values $t_{n}$ and $f_{j}$ depends on the formulation of the problem. Here, we chose frequencies varying from $f_{0}=40$ Hz to $f_{N_{freq}}=500$ Hz, with a resolution of Δf=10 Hz, given that is consistent with the GW signals that we want to detect. In addition, as we apply the WT to each cleaned strain data sample, we have discrete time values varying from $t_{0}=0$ s to $t_{N_{time}}=T_{win}-1/f_{s}$, with a resolution of $\Delta t=t_{s}=1/f_{s}$, where $f_{s}=4096$ Hz is the sampling frequency of the initial segments of strain data.

Although the size of output images is not too large, we decided to apply a resizing to reduce them from 4096×47 pixels to 32×16 pixels. Keep in mind that, as we need to analyze several thousand images, using their original size would be unnecessarily expensive for system resources. The resizing was performed by a bicubic interpolation, in which each pixel of a reduced image is a weighted average of pixels in the nearest four-by-four neighborhood.

In summary, after applying the WT and the resizing to each strain data sample, we generated an image dataset ${\left\{X_{i},y_{i}\right\}}_{i=1}^{N}$, where $X_{i}\in R^{N_{time}=32\times N_{freq}=16}$ depicts each image as a matrix of pixels (notice that we are inverting the standard notation used in image processing of height×width, because we defined our images by a 2D discrete plot of *x* axis vs. *y* axis), $y_{i}\in \left\{1,2\right\}$ the classes that we are working with, and $N_{set}\in \left\{4008,3360\right\}$, depending on whether we are using data from H1 or L1, respectively.

Figure 4 shows two representative strain date samples (one belonging to C1 and other to C2) as a time domain signal, their time-frequency representation according to the WT with a Morlet wavelet as kernel, and its resized form. Both of the samples were generated from a strain data segment recorded by the L1 detector of 4094 s at GPS time 932335616. The image sample at the right shows a GW transcient that has a variable frequency approximately between 100 Hz and 400 Hz. It is important to clarify that, before entering to CNNs, image samples are rearranged, such that they the whole dataset has a size $N_{set}\times 32\times 16\times 1$, denoting $N_{set}$ images of size 32×16 pixels using 1 channel for grayscale—sample input images that are shown in Figure 4 are in color just for illustrative purposes.

### 2.4. Resampling and White-Box Approach

In Section 1, we briefly mentioned our resampling white-box approach for dealing with intrinsic uncertainties to CNN algorithms. Subsequently, it is important to clarify subtleties and advantages of this approach.

A CNN algorithm does not input waveform templates as isolated samples, but also the distribution of these templates considering the whole training and testing datasets, as Gebhard et al. [34] critically pointed out. Indeed, in all previous works of DL applied to GW data analysis, distributions of training samples were set up as class-balanced artificially by hand, which is to say, with equal or similar number of samples for each class, despite that real occurrences of classes in recorded observational data of LIGO are very different from each other—works, such as [53,54], have reported information about BBH population from O1 and O2 runs. It is a common practice in ML and DL to draw on artificial balanced datasets to made CNN algorithms easily tractable with respect to hyperparameter tuning, choice of performance metrics, and cost missclasification. Nonetheless, when uncertainty is taken into account, the real frequency of occurrence of samples cannot be ignored, because they define how reliable is our decision criteria for classifying when an algorithm outputs a score for a single input sample. These kinds of details are very known in the ML community [55], and they need to be seriously explored in GW data analysis beyond just hands-on approaches. Deeper multidisciplinary researches are necessary for advancing in this field.

The full problem of dealing with arbitrarily imbalanced datasets, for now, is beyond of our research. Nonetheless, even though working with balanced datasets, a good starting point is to include stochasticity by resampling, which in turn we define as repeated experiments of a global *k*-fold cross-validation (CV) routine. This stochasticity is different to that usually introduced in each learning epoch by taking a mini-batch of the whole training set for updating the model parameters (e.g., by a stochastic gradient descent algorithm) and, therefore, minimizing the cross-entropy. For this research, we will consider the above two sources of stochasticity.

Stochastic resampling helps to aliviate artificiality that is introduced by a balanced dataset, because the initial splitting into *k* folds is totally random and each whole *k*-fold CV experiment is not reproducible in a deterministic fashion. Besides, this approach implemented an experimental setup in which uncertainties are even more evident and need to be seriously treated beyond of just reporting metrics of single values. Indeed, in most common situations with really big datasets (i.e., millions, billions, or more samples), stastistical tools for decreasing system resources, and data changing over time, among others; CNN algorithms are generally set such that their predictions are not deterministic, leading to distributions of performance metrics instead of single value metrics—and demanding formal probabilistic analyses. Given these distributions, with their inherent uncertainties, a statistical paired-sample test is necessary for formally concluding how close or far our CNNs are to a totally random classifier, and that we performed in this work.

On the other hand, our white-box approach works as a complementary tool to understand how uncertainties influence performance of CNN algorithms. In principle, white-box testings depend on the complex and still open problem of explainability in ML/DL. Nonetheless, for our purposes it is enough to define “white-box” as a methodology that appeals to the information regarding how the CNN algorithms internally behave. A crucial aspect here is to explicitly (mathematically) describe how each layer of a CNN works and why to choose them. This is actually the most basic explanatory procedure when considering that, from a fundamental point of view, we still do not have analytical theories to explain low generalization errors in DL or even, to establish analytical criteria to unequivocally choose a specific architecture for performing a particular task [56]—in practice, we only have a lot of previously implemented CNN architectures for other problems, which need to be test as a just essay-error process for a new problem, actually. This white-box approach allowed us to smartly choose a reduced set of hyperparameters to be adjust, which was very useful for avoiding unnecessary intensive explorations given our limited system resources. Moreover, this approach assisted us in understanding why the resulting performances of our classifications are class- and threshold-dependent based on the distribution of output probabilistic scores.

### 2.5. CNN Architectures

Our CNN algorithms consist of two main stages in terms of functionality: feature extraction and classification. First stage begins inputting images from a training dataset and the second stage ends ouputing predicted classes for each image sample. Classification is our ultimate goal, consisting in a perceptron stage plus an activation function as usual in ANNs. Feature extraction, on the other and, is the core of CNNs, because it provides the ability of image recognition, consisting in three substages: convolution, detection, and pooling. These substages are implemented through layers that, as a whole, define a stack. Subsequetly, this stack can be connected to a second stack, and so on, until last stack is connected to the classifier. For this work, we tested CNNs with 1, 2, and 3 stacks.

Knowing that general functionality of our CNNs is a must, but, for contributing with a white-box approach, we need to understand what each layer does. For this reason, we proceed to mathematically describe each kind of layer that were used—not only those that are involved in the already mentioned (sub)stages of the CNN, but also those that were required in our hands-on implementation with the MATLAB Deep Learning Toolbox [57]. Take in mind that the output of a layer is the input of the next layer, as detailed in Figure 5 for a single-stack CNN.

Image Input Layer. Inputs images and applies a zero-center normalization. Denoting an i−th input sample as the matrix of pixels $X_{i}\in R^{N_{time}\times N_{freq}}$ belonging to a dataset of $N_{train}$ same size training images, this layer outputs the normalized image:
(8)$$X^{norm}{}_{i}\left(j,k\right)=X_{i}\left(j,k\right)-\frac{1}{N_{train}}\sum_{i=1}^{N_{train}}X_{i}\left(j,k\right)\;,$$
where the second term is the average image of the whole dataset. Normalization is useful for dimension scaling, making changes in each attribute, i.e., each pixel (j,k) along all images, of a common scale. Because normalization does not distort relative intensities too seriously and helps to enhance contrast of images, we can apply it to the entire training dataset, independently what class each image belong for.Convolution Layer. Convolves each image $X^{norm}{}_{i}$ with $C_{K}$ sliding kernels of dimension $K_{time}\times K_{freq}$. Denoting each l−th kernel by $K^{l}$ with $l\in \left\{1,2,...,C_{K}\right\}$, this layer outputs $C_{K}$ feature maps, and each of them is an image that is composed by the elements or pixels:
(9)$$M^{l}{}_{i}\left(p,q\right)=\sum_{m}\sum_{n}X^{norm}\left(m,n\right)K^{l}\left(p-m+1,q-n+1\right)+b\;,$$
where *b* is a bias term, and indices *p* and *q* run over all values that lead to legal subscripts of $X^{norm}\left(m,n\right)$ and $K^{l}\left(p-m+1,q-n+1\right)$. Depending on the parametrization of subscripts *m* and *n*, dimension of images $M^{l}{}_{i}$ can vary. If we include the width $M_{time}$ and height $M_{freq}$ of output maps (in pixels) in of a two-dimensional vector just for notation, these spatial sizes are computed by:   
(10)$$\begin{array}{c} \left(M_{time},M_{freq}\right)=\frac{1}{\left(str_{time},str_{freq}\right)}\left[\left(N_{time},N_{freq}\right)-\left(K_{time},K_{freq}\right)\right.\\ +\left.2(pad_{time},pad_{freq})\right]\left(1,1\right)\;, \end{array}$$
where str (i.e., stride) is the step size in pixels with which a kernel moves above $X^{norm}{}_{i}$, and padd (i.e., padding) denotes time rows and/or frequency columns of pixels that are added to $X^{norm}{}_{i}$ for moving the kernel beyond the borders of the former. During the training, components of kernel and bias terms are iteratively learned from certain initial values appropriately chosen (see Section 2.6); then, once the CNN has captured and fixed optimal values for these parameters, convolution is applied to all testing images.ReLU Layer. Applies the Rectified Linear Unit (ReLU) activation function to each neuron (pixel) of each feature map $M^{l}{}_{i}$ obtained from the previous convolutional layer, outputting the following:
(11)$$R^{l}{}_{i}\left(p,q\right)=\max\left\{0,M^{l}{}_{i}\left(p,q\right)\right\}\;.$$In practice, this layer detects nonlinearities in input sample images; and, its neurons can output true zero values, generating sparse interactions that are useful for reducing system requirements. Besides, this layer does not lead to saturation in hidden units during the learning, because its form, as given by Equation (Equation 11), does not converge to finite asymptotic values. (Saturation is the effect when an activation function located in a hidden layer of a CNN converge rapidly to its finite extreme values, becoming the CNN insensitive to small variations of input data in most of its domain. In feedforward networks, activation functions as sigmoid or tanh are prone to saturation, hence they use are discouraged except when the output layer has a cost function able to compensate their saturation [23] as, for example, the cross-entropy function).Max Pooling Layer. Downsamples each feature map $R^{l}{}_{i}$ with the maximum on local sliding regions $L_{R}$ of dimension $P_{time}\times P_{freq}$. Each pixel of a resulting reduced featured map $m^{\ell }{}_{i}$ is given by the following:
(12)$$m^{l}{}_{i}\left(r,s\right)=\max_{r,s}\left\{R^{l}{}_{i}\left(p,q\right)\right\}\;,\;\;\forall \left(p,q\right)\in L_{R}\;,$$
where ranges for indices *r* and *s* depend on the spatial sizes of outputs maps; and these sizes, i.e., width $m_{time}$ and height $m_{freq}$, being included in a two-dimensional vector just for notation, are computed by:
(13)$$\begin{array}{c} \left(m_{time},m_{freq}\right)=\frac{1}{\left(str_{time},str_{freq}\right)}\left[\left(M_{time},M_{freq}\right)-\left(P_{time},P_{freq}\right)\right.\\ \left.+2(pad_{time},pad_{freq})\right]+\left(1,1\right)\;, \end{array}$$
where the padding and stride values have the same meanings as in the convolutional layer. Interestly and apart of reducing system requeriments, max pooling layer leaves invariant output values under small translations in the input images, which could be useful for working with a network of detectors—the case in which a detected GW signal appears with a time lag between two detectors.Fully Connected Layer. This is the classic perceptron layer used in ANNs and it performs the binary classification. It maps all images $m^{\ell }{}_{i}$ to the two-dimensional vector $h_{i}$ by the affine transformation:
(14)$$h_{i}=W^{T}m^{flat}{}_{i}+b\;,$$
where $m^{flat}{}_{i}$ is a vector of $N_{fc}$ dimensions, with $N_{fc}$ the total number of neurons considering all input feature maps, b a two-dimensional bias vector, and W a weight matrix of dimension $2\times N_{fc}$. Similarly to the convolutional layer, elements of W and b are model parameters to be learn in the training. Matrix $m^{flat}{}_{i}$ includes pixels of all feature maps $m^{l}{}_{i}$ (with $l=1,2,...,C_{K}$) as a single “flattened” column vector of pixels; then, information about topology or edges of sample images is lost.Softmax Layer. Applies the softmax activation function to each component *j* of vector $h_{i}$:
(15)$$y^{j}{}_{i}=softmax\left(h^{j}{}_{i}\right)=e^{h^{j}{}_{i}}/\sum_{j}e^{h^{j}{}_{i}}\;,$$
where j=1,2, depending on the class. Softmax layer is the multiclass generalization of sigmoid function, and we include it in the CNN, because, by definition, transform real output values of fully connected layer in probabilities. In fact, according to [58], output values $y^{j}{}_{i}\in \left[0,1\right]$ are interpreted as posterior distributions of class $c^{j}$ conditioned by model parameters. That is to say $y^{j}{}_{i}\left(\theta \right)=P_{i}\left(c^{j}|\theta \right)$, where θ is a multidimensional vector containing all model parameters. It is common to refer to $y^{j}{}_{i}\left(\theta \right)$ values as the output *scores* of the CNN.Classification Layer. Stochastically takes $\tilde{N}<N_{train}$ samples and computes the cross-entropy function:
(16)$$\begin{array}{ccc} E\left(\theta \right) & = & -\ln L\left(\theta |y^{1}{}_{i},y^{2}{}_{i}\right)=-\ln\prod_{i=1}^{\tilde{N}}\underset{y^{1}{}_{i}}{\underset{︸}{P_{i}\left(c^{1}|\theta \right)}}\underset{y^{2}{}_{i}}{\underset{︸}{P_{i}\left(c^{2}|\theta \right)}}\\ & = & -\sum_{i=1}^{\tilde{N}}\left(\ln y^{1}{}_{i}+\ln y^{2}{}_{i}\right)\;, \end{array}$$
where $y^{1}{}_{i}$ and $y^{2}{}_{i}$ are the two posterior probabilites that are outputted by softmax layer and L a likelihood function. Cross-entropy is a measure of the risk of our classifier and, following a discriminative approach [22], Equation (Equation 16) defines the maximum likelihood estimation for parameters included in θ. Now, we need now a learning algorithm for maximizing the likelihood $L\left(\theta |y^{1}{}_{i},y^{2}{}_{i}\right)$ or, equivalently, minimizing $E\left(\theta \right)$, with respect to model parameters. Section 2.6 introduces this algorithm. (Take in mind that our approach estimates the model parameters through a feedforward learning algorithm from classification layer to previous layers of the CNN. Alternatively, when considering that posterior probability outputted by softmax layer is $P_{i}\left(c^{j}|\theta \right)=P_{i}\left(\theta |c^{j}\right)P\left(c^{j}\right)/P\left(\theta \right)$ because of the Bayes theorem, the other approach could be maximize likelihood funcion $P_{i}\left(\theta |c^{j}\right)$ with respect to model parameters. This alternative approach is called generative and will be not considered in our CNN model. In short, in Section 3.4 we will present the simplest generative models to compare with our CNN algorithms, namely Naive Bayes classifiers).

Table 1 shows the three architectures that we implemented for this study. A standard choice for convolutional kernel size is 5×5, but, as we need to recognize images of width greater than height, we set a first kernel with size 5×4. Besides, we chose a size 2×2 for a first max pooling region, which is also an standard downsampling option. Subsequently, with these dimensions, we have that our CNN algorithms can increase the number of stacks only to 3, where the minimum output layer size is reached—provided that the size of kernel and max pooling regions becomes smaller if they are located in subsequent stacks. This scenario is very convenient, because it allowed us to importantly decrease system resources in our hyperparameter adjustments, as will seen in Section 3.2. For choosing the number of kernels $C_{k}$, there is not a beforehand criteria, then we leave it as a matter of experiments.

### 2.6. Model Training

As detailed, model parameters are included in vector θ, which, in turn, appear in the cross-entropy function (Equation 16). Starting from given initial values, these parameters have to be learned by an minimization of the cross-entropy, taking $\tilde{N}<N_{train}$ random image samples, i.e., a mini-batch. For model parameters updating, we draw on the known gradient descent algorithm, including a momentum term to boost iterations. Denoting model parameters at the r−th iteration or epoch as $\theta^{\left(r\right)}$, then its updating at the (r+1)−th iteration is given by the following optimization rule:(17)$$\theta^{\left(r+1\right)}=\theta^{\left(r\right)}-\alpha \nabla_{\theta^{\left(r\right)}}E\left(\theta^{\left(r\right)}\right)+\gamma \left(\theta^{\left(r\right)}-\theta^{\left(r-1\right)}\right)\;,$$
where we have that $\nabla_{\theta^{\left(r\right)}}$ is the gradient with respect to model parameters and *E* the cross-entropy, in addition to two empirical quantities to be set by hand, namely the learning rate α and momentum γ. Given that we are computing the cross-entropy with $\tilde{N}<N_{train}$ random samples, the above rule is called the mini-batch stochastic gradient descent (SGD) algorithm. Besides, for all our experiments, we set a learning rate α=1.00, a momentum γ=0.9, and a mini-batch size of $\tilde{N}=128$ image samples.

With regard to the initialization of weights, we draw on Glorot initializer [59]. This scheme independently samples values from a uniform distribution with a mean equal to zero and a variance given by $2/\left(n_{in}+n_{out}\right)$, where $n_{in}=K_{time}K_{freq}$ and $n_{out}=K_{time}K_{freq}C_{K}$ for convolutional layers, and $n_{in}=C_{K}\;size\left(m_{i}^{flat}\right)$ and $n_{out}=2C_{K}$ for the fully connected layer—remember that $C_{K}$ is the number of kernels of dimension $K_{time}\times K_{freq}$ and $m_{i}^{flat}$ the vector that is inputted by the fully connected layer. All biases, on the other hand, are initialized with zeros.

Finally, whether we work with data from H1 or L1 detector, $N_{train}$ will be significantly greater than $\tilde{N}=128$; nevertheless, its specific value depends on our global validation technique is explained in Section 2.7.

### 2.7. Global and Local Validation

For this research, we only used real LIGO strain data with a given and limited number of CBC hardware injections, removing beforehand the instrumental freedom to generate an arbitrarily diverse bank of GW templates based on software injections with numerical and/or analytical templates. Under this approach, we intentionally adopt a limitation in which we cannot generate big datasets, and global validation techniques that are based on resampling are required to reach good statistical confidence and perform fair model evaluations. Even when using syntetic data from numerical relativity, and because of current system resources limitations [60], the order of magnitude of generated templates still is quite small when compared to the big data regime [61] with volumes of petabytes, exabytes, or even more; the aforementioned techniques are yet to be required. For these reasons, we implemented the *k*-fold CV technique [62], that consists in the following recipe: (i) split the original dataset into *k* nonoverlapping subsets to perfom *k* trials, then (ii) for each i−th trial use the i−th subset for testing, leaving the rest of the data for training, and finally (iii) compute the average of each performance metric across all trials.

It is known that the value of *k* in *k*-fold CV defines a trade-off between bias and variance. When k=N (i.e., leave-one-out cross-validation), the estimation is unbiased, but variance can be high and, when *k* is small, the estimation will have lower variance, but bias could be a problem [63]. For several classic ML algorithms, previous works have suggested that k=10 represents the best trade-off option ([64,65,66]), then we decided to take this value as a first approach. Moreover, in the following works [67,68], we decided to perform 10 repetitions of the 10-fold cross-validation process [55], in order to reach a good stability of our estimates, to present fair values of the cross-entropy function given the stochastic nature of our resampling approach, and, more important, obtain information regarding the distribution of accuracy (and other standard metrics) in which there is involved uncertainty.

Moreover, *k*-fold CV helps to aliviate the artificiality that is introduced by balanced dataset, because the initial splitting into k folds is totally random. In research [34] authors warns about the fact that CNNs not only capture GW templates alone, but also transfer to the test stage the exact same probability distribution given in the training set. This claim is true but, it is important to take in mind that, working with balanced datasets, as a first approach, is simply motivated by the fact that many of the standard performance metrics give excessive optimistic results on classes of higher frequency in imbalanced dataset, and dealing with arbitrarily unbalanced datasets is not a trivial task. In any case, including *k*-fold CV as a random resampling, starting from a balanced dataset, is desirable for statistical purposes.

With the sizes of our datasets detailed at the end of Section 2.3.3 and the 10-fold CV, we have a training set of $N_{test}=floor\left[4008/10\right]=400$ samples obtained from H1 detector, and of $N_{test}=floor\left[3360/10\right]=336$ samples from the L1 detector. Consequently, $N_{train}=3607$ for H1 data and $N_{train}=3607$ for L1 data.

Local validation, which is to say validation performed within a learning epoch, is also an crucial ingredient of our methodology. In particular, our algorithm splits the training set of $N_{train}$ data into two subsets, one for the training itself ($0.9N_{train})$ and the other for validation ($0.1N_{valid}$). Validation works as a preparatory mini test that is useful for monitoring learning and generalization during the training process.

With regard to regularization techniques, local validation was performed once per $floor[N_{train}/\tilde{N}]$ epochs and cross-entropy was monitored with a validation patience p=5 (value given by hand), which simply means that, if $E^{\left(r+1\right)}\geq E^{\left(r\right)}$ occurs *p* times during the validation, then the training process is automatically stopped. Besides, to avoid overfitting, training samples were randomized before training and before validation, solely in the first learning epoch. This randomization is performed, such that the link between each training image $X_{i}$ and its respective class $y_{i}$ is left intact, because we do not want to “mislead” our CNN when it learns from known data.

### 2.8. Performance Metrics

Once our CNN is trained, the goal is predicting classes of unseen data, i.e., data on which the model was not trained and, of course, achieve a good performance. Hence, performance metrics in the test are especially crucial. When considering the last layer of our CNNs, a metric that is natural to monitor during the training and validation process is the cross-entropy. Other metrics that we use come from counting predictions. As our task has to do with a binary classification, these metrics can be computed from the elements of a 2×2 confusion matrix, namely true positives (TP), false positives (FP) or type I errors, true negatives (TN), and false negatives (FN) or type II errors, as detailed in Table 2.

Accuracy is the most used standard metric in binary classifications. Besides, all of the metrics shown in the right panel of Table 2 depend on a choosen threshold as a crucial part our decision criterion. Depending of this threshold, and the output score for an input image sample, our CNN algorithms assign class 1 or class 2. Although the threshold is fixed for all these metrics, one can also vary it for generating the well known Receiving Operating Characteristic (ROC) [69] and Precision-Recall [70] curves, among others. The first curve describes the performance of the CNN in the fall-out (or false positive rate) vs. recall (or true positive rate) space, and the second one in the recall vs. precision space. Given that the whole curves are generated by varying the threshold, each point of those curves represents the performance of the CNN given a specific threshold. As we worked with balanced datasets, ROC curves are the most suitable option. In particular, for a probabilistic binary classifier, each point of its ROC curve is given by the ordered pair:(18)$$ROC_{thr}={\left.\left(Fall-out,Recall\right)\right|}_{thr}\;\;,$$
where 0<thr<1. Finally, F1 score and G mean1 are metrics that summarize in a single metric pairs of other metrics and being, in general, useful for imbalanced multiclass classifications. In short, we decided to compute these two last metrics, because they give useful moderation features for performance evaluation—more details are presented in Section 3.3.

We also want to perform a shuffling in order to ensure that our results are statistically significant. Our algorithm already performs a randomization over the training set, before the training and before the validation in order to prevent overfitting, as mentioned in Section 2.7. However, the shuffling applied here is more radical, because it broke the link between training images and their respective classes, and it is made by random permutation over indices *i* solely for the matrices $X_{i}$, belonging to training set ${\left\{X_{i},y_{i}\right\}}_{i=1}^{N_{train}}$, before each the training. Subsequently, if this shuffling is present and our results are truly significant, we expect that accuracy in testing be lower than that computed when no shuffling is present, reaching values around 0.5—as this is the chance level for a binary classifier. This will be visually explored, looking at the dispersion of mean accuracies in boxplots and, more formally, confirmed by a paired-sample t-test of statistical inference. For each distribution of mean accuracies, with and without shuffling, we can define sets D and $D_{shuff}$, respectively. Subsequently, with means of each of these sets, namely μ and $\mu_{shuff}$, the goal is testing the null hypothesis $H_{0}:\mu -\mu_{shuff}=0$, and, then, we just would need to compute the p-value, i.e., the probability of resulting accuracies be possible assuming that the mentioned null hypothesis is correct, given a level of significance—Section 3.5 presents more details about shuffling and consequent statistical tests.

## 3. Results and Discussion

### 3.1. Learning Monitoring per Fold

While the mini-batch SGD was running, we monitored the cross-entropy and accuracy evolution along epochs. This is the very first check to ensure our CNNs were properly learning from data and our local validation criteria stopped the learning algorithm in the right moment. If our CNNs were correctly implemented, we expected that cross-entropy be minimized to reach values as close as possible to 0, and the accuracy to reach values as close as possible to 1. If this check gave wrong results, then there would be no point in computing subsequent metrics.

Figure 6 shows two representative examples of this check, using data from H1 and L1 detectors. Both were performed during a single fold of a 10-fold CV experiment, from the first to the last mini-batch SGD epoch. Besides, here we used a time resolution of $T_{win}=0.50$ s with 2 stacks, and 20 kernels in convolutional layers. Notice that cross-entropy shows decreasing trends (Figure 6a,c) and accuracy increasing trends (Figure 6b,d). The total number of epochs for H1 data was 372, and for L1 data was 513, which means that the CNN has greater difficulties in learning parameters with L1 than with H1 data. When the CNN finish its learning process, cross-entropy and accuracy reach values of 0.184 and 0.945, respectively, using H1 data; and 0.357 and 0.805, respectively, using L1 data.

Notice that, from all plots shown in Figure 6, fluctuations appear. This is actually expected, since, in the mini-batch SGD algorithm, a randomly number of samples $\tilde{N}<N_{train}$ are taken, then stochastic noise is introduced. Besides, when using data from L1 detector, some anomalous peaks appear between epoch 350 and 400, but this is not a problem because CNN normally continues its learning process and trendings in both metrics are not affected. At the end, we can observe this resilience effect, because of our validation patience criterion, which is implemented to prevent our CNN algorithm prematurely stopping and/or to dispense with manually adjust the total number of epochs for each learning fold.

Still focusing on the SGD fluctuations, zoomed plots in Figure 6 show their order of magnitude—the highest peak minus the lowest peak. When we work with H1 data, cross-entropy fluctuations are about 0.130 (Figure 6a) and accuracy fluctuacions are about 0.090 (Figure 6b). On the other hand, when we learn from L1 data, both cross-entropy and accuracy fluctuations are about 0.080 (Figure 6c,d). Here, it should be stressed that, although mini-batch SGD perturbations contribute with its own uncertainty, when we compute mean accuracies among all folds in the next Section 3.2, we will see that the magnitude of these perturbations do not totally influence the magnitude of data dispersion present in the distribution of mean accuracies.

### 3.2. Hyperparameter Adjustments

Our CNN models introduce several hyperparameters, namely the number of stacks, size and mount of kernels, size of pooling regions, number of layers for each stack, stride, and padding, among many others. Presenting a systematic study for all hyperparameters is beyond the scope of our research. However, given CNN architectures shown in Table 1, we decided to study and adjust two of them, namely the number of stacks and the number of kernels in convolutional layers. In addition, although it is not an hyperparameter of the CNN, the time length $T_{win}$ of the samples is a resolution that also required being set to reach an optimal performance, then we included it in the following analyses.

A good or bad choice of hyperparameters will affect the performance, and this choice introduces uncertainty. Nonetheless, once we found a set of hyperparameters defining an optimal setting and, regardless of how sophisticated our adjustment method is, the goal is only using this setting for predictions. That is to say, once we find optimal hyperparameters, we remove the randomness that would be introduced in predictions if we run our CNN algorithms with different hyperparameters. For this reason, we say that uncertainty that is present in hyperparameter selection defines a prior level to intrinsic uncertainties of a setting already chosen—according to the Bayesian formalism, this is a prior belief. Last uncertainties are more risky, because they introduce stochasticity in all predictions of our models, despite that we are working with a fixed particular hyperparameter setting.

In any case, we emphasize that our methodology for hyperparameter selection is robust. Thanks to our white-box approach, a clear understanding of the internal behavior of our CNNs was shown and, based on this understanding, we heuristically proposed a reduced set of hyperparameters to perform a transparent statistical exploration on possible meaningful values—obtaining good results, as will be seen. This methodology is smartly distancing from the blind brute force approach of performing unnecesarily large and expensive explorations (many hyperparameter settings could be silly to waste time on them), and even much more far from a naive perspective of trivializing model decision as a superficial detail and/or something opaquely given without showing a clear exploration.

The first hyperparameter adjustment is shown in Figure 7, and it was implemented to find optimal number of stacks and time length resolution, according to the resulting mean accuracies. The left panels (Figure 7a,c) show the distribution of mean accuracy for all 10 repetitions or runs of the entire 10-fold CV experiment, in function of $T_{win}$. Each of these mean accuracies, which we can denote as $Acc_{i}{}^{CV}$, is the average among all fold-accuracies of a i−th run of the 10-fold CV. In addition, the right panels (Figure 7b,d) show the mean of mean accuracies among all 10 runs, i.e., $\bar{Acc}=\left(1/10\right)\sum_{i=1}^{10}Acc_{i}{}^{CV}$, in function of $T_{win}$. Inside right plots, we have included small boxplots that, as will be seen next, are useful to study dispersion and skewness of mean accuracy distributions—circles inside boxplots are distribution mean values. The line plots show contributions of our three CNN architectures, i.e., with 1, 2, and 3 stacks.

Consider the top panels of Figure 7 for H1 data. Notice, from Figure 7b that, for all CNN architectures, mean of mean accuracies shows a trend to decrease when $0.75\;s\leq T_{win}\leq 2.00$ s and this decrease occurs more pronouncedly when we work with less stacks. Besides, when $0.25\;s\leq T_{win}\leq 0.75$ s, a slight increase appears, even if local differences are of the order of SGD fluctuations. In short, given our mean accuracy sample dataset, the highest mean of mean accuracies, about 0.895, occurs when $T_{win}=0.75$ s with a CNN of 3 stacks. Subsequently, to decide if this setting is optimal, we need to explore Figure 7a together with boxplots inside Figure 7b. From Figure 7a, we have that not only the mentioned setting give high mean accuracy values, but also $T_{win}=0.25$ s and $T_{win}=1.00$ s, both with 2 stacks, $T_{win}=0.75$ s with 2 and 3 stacks, and even $T_{win}=1.25$ s with 3 stacks; all these settings reach a mean accuracy that is greater than 0.9. Setting $T_{win}=1.25$ s with 3 stacks can be discarded because its maximum mean accuracy is clearly an outlier and, to elucidate what of remaining settings is optimal, we need to explore boxplots.

Here, it is crucial to assimilate that the optimal setting to choose actually depends on what specifically we have. Let us focus on Figure 7b. If the dispersion does not worry us too much and we want to have a high probability of occurence for many high values of mean accuracy, setting of $T_{win}=0.25$ s with 2 stacks is the best, because its distribution has a slighly negative skewness concentrating most of mass probability to upper mean accuracy values. On the other hand, if we prefer to have more stable estimates working with less dispersion at the expense of having a clear positive skewness (in fact, having a high mass concentration in a region that does not reach as high mean accuracy values as the range from median to third quartile in the previous setting), setting of $T_{win}=0.75$ s with 3 stacks is the natural choice. In practice, we would like to work with greater dispersions if they help to reach the highest mean accuracy values, but, as all our boxplots have similar maximum values, we decide to maintain our initial choice of $T_{win}=0.75$ s with 3 stacks for H1 data.

From Figure 7a, it can be seen that, regardless number of stacks, data dispersion in $1.25\;s\leq T_{win}\leq 2.00$ s is greater than in $0.25\;s\leq T_{win}\leq 1.00$ s, even if in the former region dispersion slightly tends to decrease as we increase the number of stacks. This actually is a clear visual hint that, together with the evident trend to decrease mean accuracy as $T_{win}$ increase, motivate discarding all settings for $T_{win}\geq 1.25$ s. However, this hint is not present in the bottom panels, in which a trend of decrease and then increase appears (which is clearer in the right plot), and data dispersion of mean accuracy distributions are similar almost for all time resolutions. For this reason, and although the procedure for hyperparameter adjustment is the same as upper panels, one should be cautious, in the sense that decisions here are more tentative, especially if we have prospect to increase the mount of data.

In any case, given our current L1 data and based on scatter distribution plot in Figure 7c and line mean of mean accuracies plot in Figure 7d, we have that the best performance(s) should be among settings of $T_{win}=0.5$ s with 2 stacks, $T_{win}=0.75$ s with 2 and 3 stacks, and $T_{win}=1.00$ s with 3 stacks. Now, exploring the boxplots in Figure 7d, we notice that, even if settings of $T_{win}=0.75\;s,1.00$ s with 3 stacks reach the highest mean accuracy values, their positive skewness toward lower values of mean accuracies is not great. SUbsequently, the two remaining settings, which, in fact, have negative skewness toward higher mean accuracy values, are the optimal options, and again choosing one or other will depend of what extent we tolerate data dispersion. Unlike upper panels, here a larger dispersion increase the probality to reach higher mean accuracy values; therefore, we finally decide to work with the setting of $T_{win}=0.75$ s with 2 stacks for L1 data.

To find the optimal mount of kernels in convolution layers, we perfomed the adjustment that is shown in Figure 8, again separately for data from each LIGO detector. When considering the information that is provided by previous adjustment, we set $T_{win}=0.75$ s, and the number of stacks in 3 for H1 data and 2 for L1 data. Subsequently, once the 10-fold CV was run 10 times as usual, we generated boxplots for CNN configurations with several mounts of kernels, as was advanced in Table 1, including all mean accuracies for each run marked (red circles). Besides, the average for each boxplot is included (blue crosses). Random data horizontal spreading inside each boxplot was made to avoid visual overlap of markers, and it does not mean that samples were obtained with a number of kernels different from those already specified in horizontal axis.

Let us concentrate on kernels adjustment for H1 data in the left panel of Figure 8. From these results we have that a CNN with 12 kernels give us more stable results by far, because most of its mean accuracies lie in the smallest dispersion region—discarding outliers, half of mean accuracies are concentrated in a tiny interquartile region located near to 0.895. On the other hand, CNN configuration with 24 kernels is the least suitable setting among all, not because its mean accuracy values are low per se (values from 0.800 to 0.898 are actually good), but rather, because, unlike other cases, the nearly zero skewness of its distribution is not prone to boost sample values beyond the third quartile as it is appeared. Configuration with 8 kernels has a distribution mean very close to the setting with 24 kernels and, even, reaches two mean accuracy values of about 0.905. Nonetheless, given that settings with 16, 20, 28, and 32 have mean of mean accuracies greater or equal to 0.895 (and, hence, boxplots that are located towards relative higher mean accuracy values), these last four configurations offer the best options. At the end, we decided to work with 32 kernels, because this setting groups a whole set of desirable features: the highest mean of mean accuracies, namely 0.893, a relatively low dispersion, and a positive skewness that is defined by a pretty small range from the first quartile to the median.

Kernels adjustment for L1 data is shown in right panel of Figure 8. Here, the situation is easier to analize, because performance differences appears to be visually clearer than those for data from H1. Settings with 8, 20, and 28 kernels lead to mediocre performances, specially the first one which has a high dispersion and 70% of its samples are below 0.8 of mean accuracy. Notice that, like adjustment for H1 data, setting with 12 kernels shows the smallest dispersion (discarding a outlier below 0.77), where we have mean accuracies from 0.805 to 0.830, and, again, this option will be suitable if we would be very interested in reaching stable estimates. We decided to pick up setting with 16 kernels, which has the highest distribution mean, 0.825, 50% of mean accuracy samples above the distribution mean, an aceptable data dispersion (without counting the clear outlier), and a relatively small region from the minimum to the median.

In summary, based on all of the above adjustments, the best time resolution is $T_{win}=0.75$ s, with a CNN architecture of 3 stacks and 32 kernels when working with data from H1 detector, and 2 stacks and 16 kernels when working with data from the L1 detector. We only use these hyparameter settings hereinafter.

Now, let us finish this subsection reporting an interesting additional result. We can ask to what extent magnitude of perturbations from the mini-batch SGD algorithm influence the dispersion of mean accuracy distributions, as was mentioned at the end of Section 3.1. Here, we can compare the order of magnitude of SGD perturbations and dispersion present in boxplots. In previous subsection, we had that, when $T_{win}=0.50$ s and we work with a CNN architecture of 2 stacks and 20 kernels in convolution layers, the order of magnitude of SGD fluctuations in accuracy is about 0.090. Curiously, this value is much greater than the dispersion of data distribution shown in the left panel of Figure 7, which, in turn, reach a value of 0.025, that is to say, 3.6 times smaller. These results are good news, because apart of showing that stochasticity of mini-batch SGD perturbations do not totally define dispersion of mean accuracy distributions, it seems that our resampling approach, actually contributes to smooth stochastic effects of mini-batch SGD perturbations and, hence, to decrease the uncertainty in the mentioned distributions. This is a very important result that could serve as motivation and standard guide to future works. Given that very few previous works of ML/DL applied to GW detection have transparently reported their results under a resampling regime, i.e., clearly showing distributions of their performance metrics (for instance, [29,39]), this motivation is highly relevant. Resampling is a fundamental tool in ML/DL that should be used, even if the involved algorithms are deterministic, because there will always be uncertainty given that data are always finite. Moreover, under this regime, it is important to report the distributions of metrics to understand the probabilistic behavior of our algorithms, beyond mere averages or single values from arbitrarily picked out runs.

### 3.3. Confusion Matrices and Standard Metrics

In general, accuracy provides information regarding the probability of a successful classification, either if we are classifying a noise alone sample (C1) or noise plus GW sample (C2); that is to say, it is a performance metric with multi-label focus. However, we would like to elucidate to what extent our CNNs are proficient in separately detecting samples of each class, then it is useful to introduce peformance metrics with single-label focus. A standard tool is the confusion matrix, which is shown in Figure 9, depending on data from each detector. As we are under a resampling regime, each element of confusion matrices is computed when considering the entire mount of $100\;N_{test}$ detections, which, in turn, are resulting from concatenating all prediction vectors of dimension $N_{test}$ that are outputted by the 10 runs of the 10-k fold CV.

A first glance of the confusion matrices shown in Figure 9 reveals that our CNNs have a better performance in detecting noise alone samples than detecting noise plus GW samples, because (C1,C1) (we are using the notation $\left(row,column\right)$ to represent each element of a confusion matrix) the element is greater than (C2,C2) for both matrices. Yet, the amount of successful predictions of noise plus GW are reasonably good because they considerably surpass a totally random performance—as described by successful detections or the order of 50% of total negative samples.

Moreover, from Figure 9, we have that, based on wrong predictions, CNNs are more likely to make a type II error than type I error, because (C2,C1)>(C1,C2) for both confusion matrices. If we think more carefully, this result leads to an advantage and a disadvantage. The advantage is that our CNN performs a “conservative” detection of noise alone samples in the sense that a sample will be not classified as beloging to class 1 unless the CNN is very sure, which is to say the CNN is quite precise to detect noise samples. Using H1 data, 8533/(8533+427)≈0.952 of samples predicted as C1 belong this class; and, using L1 data, 6075/(6075+445)≈0.932 of samples that were predicted as C1 belong this class. This is an important benefit if, for instance, we wanted to apply our CNNs to remove noise samples from a segment of data with a narrow marging of error in addition to other detection algorithms and/or analysis focused on generating triggers. Nonetheless, the disadvantage is that a not less number of noise samples are lost by wrongly classifying them as GW event samples. In terms of false negative rates, we have that 1413/(8533+1413)≈0.142 of actual noise samples are misclassified with H1 data, and 1842/(6075+1842)≈0.233 of actual noise samples are misclassified with L1 data. This would be a serious problem if our CNNs were implemented to decide whether an individual trigger is actually a GW signal and not a noise sample—either Gaussian or non-Gaussian noise.

Taking in mind that, according to statistical decision theory, there will always be a trade-off between type I and type II errors [71]. Hence, given our CNN architecture and datasets, it is not possible to reduce value of (C2,C1) element without increasing value of (C1,C2) element. In principle, keeping the total number of training samples, we could generalize the CNN architecture for a multi-label classification to further specify the noise including several kind of glitches as was implemented in works as [33,38]. Indeed, starting from our current problem, such multiclass generalization could be motivated to redistribute the current false negative counts (C2,C1) in new elements of a bigger confusion matrix, where several false positive predictions will be converted to new sucessful detections located along a new longer diagonal. Nonetheless, it is not clear how to keep constant the bottom edge of the diagonal of the original binary confusion matrix when the number of noise classes is increased; not to mention that this approach can be seen as a totally different problem instead of a generalization.

With regard to misclassified GW event samples, despite that they are quite less than misclassified noise samples, we would like to understand more about them. Subsequently, we decided to study the ability of the CNN to detect GW events depending on the values of their expected SNR—values that are provided with LIGO hardware injections. The results are shown in Figure 10; upper panel with data from H1 detector, and lower panel with data from L1 detector. Both panels include a blue histogram for actual (injected) GW events that come from the testing set, a gray histogram with GW events detected for the CNN, and the bin-by-bin discrepancy between both histograms as scatter points. As a first approach, we defined this bin-by-bin discrepancy as the relative error:(19)$$Rel\;Err\left(i\right)=\left[N_{det}\left(i\right)-N_{test}\left(i\right)\right]/N_{det}\left(i\right)\;,$$
where $N_{det}$ and $N_{test}$ are the detected GW count and injected GW count, respectively, and index *i* represent a bin. Here, we set 29 same-length bins for both histograms, starting from a lower edge SNR=9.0 to a upper edge SNR=101.8 for H1 data, and from SNR=10.0 to SNR=91.2 for L1 data, respectively. For testing histograms, the count of events comes from our $100\;N_{test}$ predictions given our resampling regime.

By comparing most bins that appear on both panels of Figure 10, we have detected that the GW count is greater the more actual injections in the testing set there are. Besides, most GW events are concentrated in a region of smaller SNR values. For H1 data, most events are in the first six bins, namely from SNR=9.0 to SNR=28.9; with 6520 actual GW events and 5191 detected GW events, representing aprox. the 72.77% and 68.78% of the total number of actual GW events and detected GW events, respectively. For L1 data, on the other hand, most of the events are in the first seven bins, from SNR=10.0 to SNR=29.6; with 5040 actual GW events and 3277 detected GW events, representing aprox. the 77.30% and 70.05% of the total number of actual GW events and detected GW events, respectively.

The above information regarding counts is relevant, but the most important results come from relative errors. From these, we have that, in both panels, a clear trend of detecting a greater percentage of actual GW events as long as those events has greater SNR values. Besides, if we focus in upper panel of Figure 10, corresponding to H1 data, we have that, in first four bins, the GW count relative errors are the greatest; beginning with −0.3589 and ending with −0.1016. Subsequently, from the fifth bin at SNR=21.80, relative errors stochastically approaches zero—indeed, a relative error exactly equal to zero is reached in 10 of 29 bins. For L1 data, as shown in lower panel, we observe a similar behavior of relative error. In the first six bins, the greatest relative errors appears; from −0.5083 to −0.0977. Next, from seventh bin at SNR=26.80, relative errors stochastically approach zero. Indeed, here we have that a relative error value exactly equal to zero is reached for the first time at smaller SNR value than with H1 data, although, once zero values begin to appear, relative errors that are further away from zero than with H1 data also appear. This last result is statistically consistent with the fact that, according to Figure 9, a negative predictive value ($NPV=\left(C2,C2\right)/\left[\left(C2,C1\right)+\left(C2,C2\right)\right]$) is smaller in the confusion matrix for H1 data than for L1 data, with NPV≈0.842 and NPV≈0.717, respectively.

It should be stressed that, in the above paragraph, we refer to trends in bin-by-bin discrepances, which stochastically approach zero as the SNR values increases. Although the binning choice could influence the fact that the above discrepances are not monothonic, the main reason behind this behavior is stochasticity of our CNN algorithms. We are counting the inputted and detected GW events, which come from repeteated 10-fold CV experiments in which the dataset is stochastically split in each experiment.

For bin-by-bin discrepancies that are shown in Figure 10, we include error bars. These are standard deviations and each of these was computed from distributions of 10 relative errors because of the 10-fold CV experiment is repeated 10 times. From the plots, we observe that standard deviations do not approach to zero as their SNR increase, meaning that stochasticity introduced in such standard deviations by our resampling cannot seem to be smoothed by selecting certain SNR values.

It is important to reiterate that, here, we applied the CNN algorithm under a realistic approach in the sense that GW events are given by the hardware injections provided by LIGO, and therefore, all SNR values are given in the strain data with no possibility to be directly handled in numerical relativity templates before software injections (in addition to SNR values, frequency of occurrence of GW events also represents an important challenge to generate a more realistic dataset emulating record of astrophysical data, even though this leads us to work with highly imbalanced datasets. Even, for a more realistic situation, we could internally describe each bin of histograms in Figure 10 as a random sampling in which the counts themselves take random values, following a distribution—indeed, this hypothesis is usually assumed to perform systematic statistical comparisons between two histograms). Additionally, this is consistent with real experimental conditions, in which the SNR values of real recorded GW signals depend solely on the nature of the astrophysical sources and the noise conditions of the detectors—aspects that obviously are not handleable during an observation run. If the CNN is able to deal with this limiting scenario beforehand, then it does not learn more than what is strictly necessary, avoiding overoptimistic results or, even, underperformance. Indeed, because of this aspect, we can transparently conclude that our CNN per se is more sensitive to stocastically detectint GW signals when SNR≥21.8 for H1 data and when SNR≥26.80 for L1 data.

Continuing with our analysis, Table 3 shows a summary of several metrics that we previously defined in Table 2—again, these metrics were computed by counting the entire mount of $100\;N_{test}$ predictions given that we repeated 10 times a 10-fold CV experiment. From the table, we have that, working with L1 data, we observe that recall has a mean value telling us that 76.8% of noise alone samples are retrieved. Given these results, if we want to have chances of recovering most noise alone samples of a segment of data on our side in order to, for instance, increase in the short-term our catalogues of glitches or to fully analyze strain data in real-time observation to filter them, this CNN could be not the best option because its sensitivity is not great. The mean recall is slightly better with H1 data, 85.8%, but not as great as to considerably improve the sensitivity. Notice, on the other hand, that mean precision and mean fall-out show that our CNN is quite precise classifying noise alone samples, because once it labels a set of samples as that, for L1 data we have that 93.2% of them are actually noise alone, and just 8.69% are GW signals. Even for H1 data the results are better, because mean precision is 95.2% and a mean fall-out is 5.34%. At the end, this disparity between recall and precision is summarized in the F1 score. For H1 data, the F1 score is 0.903 and, for L1 data, is 0.842. In both cases, the mean F1 score reaches a moderate performance with numerical values lying between values of mean recall and mean precision. Besides, although fall-out plus precision is theoretically exactly 1, here we are considering means among several stochastic realizations of theses metrics; then, summation slightly differs in 0.0189 for L1 data, and 0.00540 for H1 data.

Because the F1 score has the limitation of leaving out true negatives samples, it is recomendable to report it together with G mean1. Table 3 also shows the values for the mean of this metric, namely 0.213 for H1 data and 0.256 for L1 data. These two values are low, because, by definition, G mean1 is mainly susceptible to the sensitivity of the CNN. In fact, these results elucidate a useful feature of G mean1, namely that it is works as a warning for avoiding overoptimistic performance reports based solely on accuracy. Notice that, on the other hand, that mean of G mean1 shows a slightly better performance for L1 than for H1 data; showing that G mean1 also contributes to avoiding excesive pessimistic interpretations when accuracy, or other metrics, reach lower relative results (for a *N*-labels classification, imbalanced datasets, and N>2, accuracy has a serious risk to become a pessimistic metric, and working with single-label focus metrics would be impractical when *N* is significantly larger, because we would need *N* metrics to detail the model performance. Hence, the need of drawn on metrics as F1 score and G mean1).

Table 3 shows the dispersion of metrics. For data from a given detector (H1 or L1), we observe that standard deviations of accuracy, precision, recall, and fall-out are of the same order of magnitude. This is expected because these metrics were computed directly from the same resampling of data predictions. Besides, for H1 or L1 data, we have that the standard deviation of the F1 score is also of the same order of magnitude as other metrics. However, with G mean1, we observe a slightly smaller dispersion with L1 data than with H1 data, which is consistent with the minimal improvement reported in the mean of the G mean1. In any case, this reported improvement is actually marginal, because all other metrics report a better performance of our CNN working with data from H1 detector. In the next subsection, we give more reasons to reach this conclusion.

### 3.4. ROC Comparative Analyses

As it was mentioned in Section 2.6, all of the performance metrics that are shown in Table 2 depend on a choosen fixed threshold for assigning a class per image sample. Until now, previous analyses used a threshold of 0.5 by default; but, for generating ROC curves, it is necessary to vary this thereshold from 0 to 1 as it was pointed in Equation (Equation 18). In general, ROC curves visually show to what extent our binary CNN classifier, depending on thresholds, defines the trade-off between recovered noise alone samples and GW events samples that were wrongly classified as noise alone samples. Moreover, in the context of ML/DL techniques, ROC curves are used to contrast performances of a model learning from different datasets, or more widely, to compare performances of different models. As a case of study, and for going beyond only evaluate our CNN probabilistic classifier, here we present two ROC comparative analyses: one using H1 data and other using L1 data, where each one will contrast performances of our CNN with other two classic ML models, namely Naive Bayes (NB) and Support Vector Machines (SVM). We implemented the NB and SVM classifiers with the MATLAB Statistics and Machine Learning Toolbox [72]—for theoretical details, see [63] and/or [22].

The NB and SVM models need vectors as input, then we apply a reshaping operation: each image sample $X_{i}\in R^{N_{time}\times N_{freq}}$ is flattened in a vector $x_{i}\in R^{N_{time}N_{freq}\times 1}$ such that all columns of $X_{i}$, from the first to the last, are concatenated as a one big single column. For NB model, we assume that our train set follows a Gaussian distribution, with mean and variance obtained from to the maximum likelihood estimation (MLE). For the SVM model, on the other hand, we applied a normalization for each component of $x_{i}$ along all of the training samples, and we used a linear kernel.

Take in mind that there is not definitive criteria to generate ROC curves under the resampling regime. Subsequently, following the same approach that was taken in Section 3.3 for computing confusion matrices, we considered the whole set of $100\;N_{test}$ predictions that were made by our 10×10=100 learning-testing process. In practice, this approach avoids averaging point-by-point and helps to smooth ROC curves through increasing its number of discrete generative steps. For all ROC curves, we set $T_{win}=0.75$ s for the strain samples, and for ROC curves describing the performance of our CNN, we used the same hyperparameter adjustments for stacks and kernels that, in Section 3.2, we found are the best.

Figure 11 shows the results of our comparative analyses. Notice that, in both panels, we have that all ROC curves, in general, are quite distant from the 45-degree diagonal of totally random performance, which is fairly good. Even so, depending on the used dataset, their have different performances. When the models learn (and test with) H1 data, their performance are better than with L1 data. Now, if we focus separately on each panel, we have that, for almost all thresholds, the CNN model has the best performance, NB model the worst performance, and SVM model is in the middle. However, as it is shown in zoomed plots, we have that ROC curves in both panels have some peculiarities. In the left zoomed plot, we observe that our CNN has the best performance only until its ROC curve reaches $\left(fall-out,recall\right)=\left(0.979,0.998\right)$, because the NB classifier becomes the best and it remains that way until the end—in fact, the performance of the SVM model had already been surpassed by the performance of the NB model from the point (0.960,0.994). From what happens next, very close to the north-east edge (1,1), we should not make any strong conclusion, because we are very close to the totally random performance and, therefore, the results are mainly perturbations. In the right zoomed plot, we observe that only after the point (0.923,0.957) in the ROC space, NB model becomes better that SVM model, and the CNN classifier always has the best performance.

Notice that, on all ROC curves, a specific point have been highlighted. This is called an “optimal operating point” (OOP) and it corresponds to the particular optimal threshold (OT) in which a classifier has the best trade-off between the costs of missing noise aline samples, $cost\left(FN\right)$, against the costs of raising false noise aline detections, $cost\left(FP\right)$. In the ROC space, this trade-off is defined by isocost lines of constant expected cost: (20)$$cost_{exp}=P\left[1-Recall\right]cost\left(FN\right)+N\left[Fall-out\right]cost\left(FP\right)\;,\;\;\;\;\;\;$$
where P=TP+FN and N=FP+TN. Assuming, as a first approach, that $cost\left(FN\right)=$$cost\left(FP\right)=0.5$, then OOP is just the point lying on the ROC curve that intersects the 45 degree isocost line that is closest to the north-west corner (0,1) of the ROC plot (If $cost\left(FN\right)$ and $cost\left((FP\right)$ were different and/or the dataset were imbalanced with respect classes C1 and C2, OOP and OT would be near one of its extremes. Subsequently, in that situation, ROC analysis would be more sensitive to statistical fluctuations, making difficult to take statisfically significant decisions with respect to the class with much less detections and/or samples. This situation would require more data for dealing with the imbalance or alternative analysis as precision-recall curves or cost curves). For each ROC curve, their OOP, OT, and expected costs, are included in Table 4. Notice from this table that, for the CNN classifier, OT with H1 data are not closer to the exact fifty-fifty chance value of 0.5 than OT with L1 data, which shows that the default threshold of 0.5 is actually chosen by convention, and not because it is a limit as the performance of our classifier improves. The relative difference between OT and 0.5 has nothing to do with performance, but rather with the skewness of classes and/or cost of misclassifications. We also include the optimal expected cost that is computed with Equation (Equation 20) and define the isocost curve in which the OOP lies. Notice that smaller values of $cost_{cost}$ define isocost curves that are closer to (0,1) in the ROC space.

In general, the relative performance between models can change depending on whether their ROC curves intersect. Because of this, we would like to have a metric for summarizing, regardless of thresholds, the performance of a model in a single scalar. Here, we used the total area under the ROC curve (AUC) [73]; this is a standard metric that gives the performance of the CNN averaged over all possible trade-offs between TP predictions and FP predictions. Moreover, we can use this metric to made a final choice among all models; the best model corrresponds to the highest AUC value. In practice, we computed this metric by a trapezoidal approximation, and its results are also included in the Table 4. We have that, for both datasets, $AUC_{NB}<AUC_{SVM}<AUC_{CNN}$, which allows us to conclude that, among the three models, the CNN definitely has the best performance, followed by the SVM classifier, and finally by the NB classifier.

### 3.5. Shuffling and Output Scoring

Two related analyses were conducted to ensure that the results are statistically significant were performed, as mentioned in Section 2.8. The first one was run our CNN algorithm, including a shuffling of training samples before each training, with the peculiarity of removing links between each sample and its known label. A comparison of distribution of the mean accuracies along all runs of the 10-fold CV experiment, with and without shuffling, is shown in Figure 12—remember that each point of the boxplots, i.e., a i−th mean accuracy or $Acc_{i}{}^{CV}$, as was defined Section 3.2, coming from the i−th run of the whole 10-fold CV. From this plot, we have that shuffling radically affects the results. Whether we work with data from H1 detector or L1 detector, and if shuffling is present, the distribution of mean accuracy moves towards lower values and increases its dispersion. With H1 data, the mean of mean accuracies decreases from 0.897 to 0.494 and standard deviation increases from 0.564 o 1.34; and, with L1 data, mean of mean accuracies falls from 0.825 to 0.489 and standard deviation grows from 1.98 to 2.73.

Moreover, whether shuffling is present or not, the boxplot of mean accuracies has positive or negative skewness, respectively. This makes sense, because, without shuffling, the higher mean accuracies, the greater the effort of the CNN for reaching performances with those accuracies; there is not free lunch and we expect to have a higher concentration of samples below the median than above the median and, therefore, a positive skewness. On the other hand, if we have shuffling, we know from the basics of probability that adding new points, one-by-one, to the mean accuracy distribution, is actually a stochastically symmetric process around 0.5—the theoretical limit if we have an infinite number of points in the distribution, i.e., an infinite number of runs of the *k*-fold CV experiment. Subsequently, given that, here, we obtained medians that were slightly below of 0.5 (0.499 with H1 data and 0.492 with L1 data), it is expected that there is a higher concentration of points above medians and, then, boxplots with negative skewness; because this works as a balance to maintain the symmetry of stochastic occurrences (i.e., boxplot points) around the 0.5 mean accuracy value.

Descriptive statistics is a reasonable analysis, but, to make a formal conclusion regarding the significance of our results, we performed a sample-paired t-test. Therefore, we first define the mean accuracy datasets
(21)$$D={\left\{Acc_{i}{}^{CV}\right\}}_{i=1}^{10}\;,\;D_{shuff}={\left\{Acc_{i}{}_{i}^{CV,shuff}\right\}}_{i=1}^{10}\;,$$
without and with shuffling, respectively. Subsequently, with the means of each dataset at hand, μ and $\mu_{shuff}$, the task is test the null hypothesis $H_{0}:\mu -\mu_{shuff}=0$ by computing the p−value, which is defined as:(22)$$p=\frac{\mu -\mu_{shuff}}{\sigma /\sqrt{N_{D}}}\;.$$

Subsequently, assuming a significance level α=0.05 (a standard similarity threshold between D and $D_{shuff}$), we have that: i) if p>α=0.05, then we accept $H_{0}$, or ii) if p<α=0.05, then we reject $H_{0}$. The results for p−values are shown in Figure 12, namely $2.7482\times 10^{-14}$ with H1 data and $5.6119\times 10^{-10}$ with L1 data. These values are much less than α=0.05; hence, we reject null hypothesis and conclude that, for a significance level of 5% (or confidence level of 95%), distribution D is significantly different from $D_{shuff}$. This is actually a quite good result.

As final analysis, we focus on output scoring of the CNNs. Our CNNs output scores that are probabilities generated by the softmax layer, as explained in Section 2.5. After the training, these probabilities are defined by our classes, $c^{1}$ (noise alone) and $c^{2}$ (noise plus GW), conditioned by model parameters within vector θ once they have already be learned; namely, $y^{j}\left(\theta \right)=P\left(c^{j}|\theta \right)$ (with j=1,2) for each input image sample. Histograms describing the distribution of these probabilities, considering all our $100\;N_{test}$ predictions, are included in Figure 13—all of the histograms were made using 28 same-length bins. Here, we have important results.

Firstly, in both panels of Figure 13, we have that the distribution for $y^{1}$ and distribution for $y^{2}$ are multimodal, and each one has three different modes or maximums. In addition, we observe that both of the distributions are asymmetric. Given a multimodal distribution, there are not a univocal definition of its center; it can be its mean, its median, its center of probability mass, among others. Here, we decided to define the center of distribution for as the optimal threshold (OT), because this metric is directly related to our decision criteria for assigning a class to the output score. The closer to the OT a probabilistic occurence is located, the greater uncertainty for taking a decision about what class a CNN actually is predicting with that probability. The OT values were already computed and presented in Table 4, and we included them in panels of Figure 13 as dashed lines.

For $y^{1}$ probability, we have that, in the left-hand side of OT, there is a low concentration of occurrences until before left edge bin, $\left[0,0.0360\right]$, having the greatest mode of the whole distribution; this edge bin has 0.31 of all occurrences counted along all bins if we work with H1 data, and 0.21 of all occurrences with L1 data. In contrast, in the right-hand side of OT, we have more dispersed occurrences around the two remaining modes—with H1 data, one of these reimaining modes is located at edge right bin $]0.9720,1.0080]$, and with L1 data, no mode is located at edge right bin. The distribution of $y^{2}$ is similar to that of $y^{1}$, except now it is inverted along horizontal axis, then the highest mode is at the right edge bin, two remaining modes at the left-hand side of OT, among others. For $y^{2}$, the fraction of counted occurrences in right edge bin is also the same, 0.31 with H1 data and 0.21 with L1 data.

The above results actually mean that, given our datasets, our CNNs are more optimistic predicting GW samples than predicting noise alone samples or, equivantly, more pessimistic predicting noise alone samples than predicting GW events. Hence, even though our input datasets are exactly class-balanced, predictive behavior of our CNNs is highly class dependent. Under a frequentist approach, the asymmetric shape of distributions for $y^{1}$ and $y^{2}$ is the statistical reason why the CNNs have a high precision and a not negligible false negative rate or, said more simply, why the CNNs are more “conservative” classifying samples as noise alone, than classifying as GW events—and this is coherent with that we interpreted from confusion matrices shown in Figure 9.

It is also important to notice how, depending on data, the distribution of occurrences, either for $y^{1}$ or $y^{2}$, change. Remember that, from previous ROC comparative analyses, we found that working with H1 data reaches a better performance than working with L1 data and, here, we also observed this improvement from another point of view. Because of the more uncertainty that our CNNs have for predicting a specific class, occurrences are more concentrated (i.e., skewed) towards OT. Even if our network was not learn anything, e.g., because of a shuffling, as we previously applied, then we would have that all probabilities are distributed as a Gaussian centered at the default threshold, 0.5, i.e., a totally random performance—although it is not explicitly included here, we checked this random values with our code and visualizations.

## 4. Conclusions

In this work, we applied CNN algorithms for the detection of GW signals from CBCs, through a binary classication with samples of non-Gaussian noise only and samples of non-Gaussian noise plus GW injections. Being a crucial part of the data pre-processing, we applied a Morlet wavelet transform, to convert our time series vectors (i.e., strain data) in time-frequency matrices (i.e., image data). Additionally, the resulting images were automatically decoded in the convolutional stacks of our CNNs. Besides, images in time-frequency representation were reduced in such a way that all of our CNNs were easily run in a single local CPU, reaching good performance results.

The significant novel contribution of our work is adopting a resampling white-box approach, being motivated by the need to advance towards a statistically informed understanding of uncertainties that are involved in CNNs. Moreover, as a manner to reproduce a more realistic experimental setting for testing our CNN algorithms, we draw on single-interferometer data from LIGO S6 run, considering raw strain data with noise and hardware injections of GW signals solely; that is to say, we removed the instrumental freedom of generating distributions of simulated GW signals as intensely and/or frequently as one would want.

Because of introducing stochasticity by repeated 10-fold CV experiments, almost all of the tasks were more complex than in a simple deterministic fashion, but it also forced us to acquire useful tools to overcome too much optimistic or too much pessimistic evaluations of CNN algorithms. Hyperparameter adjustments required careful interpretations of mean accuracy distributions. We tested several CNN architectures and found two that achieve optimal peformances, one with 3 stacks and 32 kernels for data from H1 detector, and other with 2 stacks and 16 kernels for data from L1 detector, and in both cases with a resolution $T_{win}=0.75$ s. Besides, we found that stochasticity introduction by mini-batch SGD in accuracies is smoothed by the resampling, which is achieved by reducing the perturbations in a factor of 3.6. This result serves as recommendation for future works to run CNNs in the resampling regime.

From analyses of confusion matrices and standard metrics, we found that whether working with H1 data or L1 data, CNN algorithms are quite precise to detect noise samples, but are not sensitive enough to recover GW signals. These results mean that, in the context of their two-label predictions, CNN algorithms are better suitable for noise sample detection than the generation of GW triggers—the results that can serve as motivation to explore alternative strategies in which CNNs cab be used as complementary pipelines for noise detection and be systematically compared with current noise reduction techniques used in LIGO pipelines. However, these conclusions regarding noise detection are not totally definitive, because if we considering only GW events of SNR≥21.8 working with H1 data, and SNR≥26.8 working with L1 data, our CNN algorithms actually could be considered as a tentative algorithm for GW detection. Here, we stress the label *tentative*, because the mentioned conditions for SNR values actually depend on the nature of initial datasets that are inputted by the CNNs that, in our case, are still class-balanced. At this point, it is evident that more research is necessary to begin dealing with arbitrarily class-imbalanced data.

With ROC curves, we also compared the CNNs with other two classic ML models, NB and SVM, reaching that CNNs have much better performances than the mentioned classic ML models—a result that is consistent with what has been reported in recent years in most of the works of DL applied to GW detection. From ROC analyses, we also found optimal thresholds, which are very useful parameters in establishing a statistical decision criterion for the classification, namely 0.430 for H1 data and 0.472 for L1 data.

In order to elucidate whether our predictions are statistically significant, we performed a paired-sample t-test, obtaining that the performace of CNN algorithms is significantly different to that of a totally random classifier, with a confidence level of 95%. Finally, thank to our white-box approach, we found that probabilistic scores are asymmetrically distributed, implying that the predictive behavior of the CNNs is highly class dependent. Here, we concluded that the discrepancy between precision and sensitiviy of the CNNs can be statistically explained by the nature of the score distributions, which, in turn, come from mathematical formulation of the softmax activation function that we included as penultimate layer in the classification stage of the CNNs. This conclusion implies that softmax layer works not only as tool to generate probabilistic scores useful for the binary classification, but also to measure the uncertainties of the CNNs given the datasets.

We presented a detailed cross-disciplinary exploration to seriously deal with the uncertainties of CNN algorithms. In particular, we show that, for achieving this goal, it is fundamental to have clear statistical information regarding the advantages and disadvantages of CNN algorithms. In general, we highly recommend that future works focused on testing DL algorithms in GW analysis should focus on the problem of how to establish more realistic experimental settings in their metodologies, and how to deal with difficulties that arise from these settings, paying greater attention to uncertainties. At the end, this is one of the main pathways to claim that DL techniques are real alternatives to standard GW detection algorithms.

## Figures and Tables

**Figure 1 sensors-21-03174-f001:**
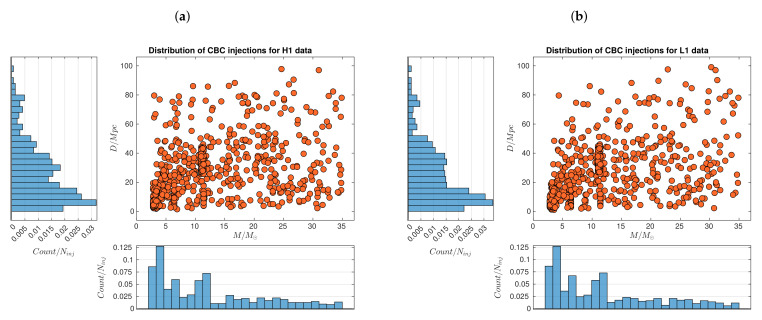
Distributions of hardware injections simulating GW emitted by CBCs that are present in S6 data from H1 (**a**) and L1 (**b**) LIGO detectors. The total masses $M=m_{1}+m_{2}$ in solar mass units and distances *D* from the sources in megaparsecs are shown. Depending on *M* and *D*, we found are several regions in which GW events are more or less frequenct. In any case, the clearest trend in data is that, as distance *D* increase from 4 Mpc, GW events are less frequent. Data dowloaded from the LIGO-Virgo GWOSC: https://www.gw-openscience.org (accessed on 1 April 2019).

**Figure 2 sensors-21-03174-f002:**
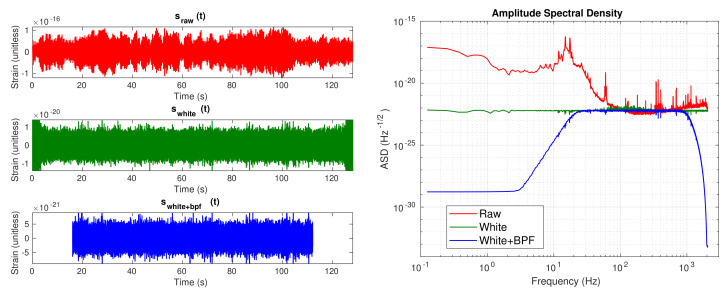
Illustration of the raw strain data cleaning. The left panel shows three plots as time series segments. First, a segment of raw strain data of 128 s after we applied the blackman window. Second, the resulting strain data after the whitening, in which it is noticeable the spurious effects at the edges. Finally, the cleaned segment of data after applied the band-pass filtering and edges removal to the previous whitened segment of data. In the right panel, we plot the same data, but now in the amplitude spectral density (ASD) vs. frequency space.

**Figure 3 sensors-21-03174-f003:**
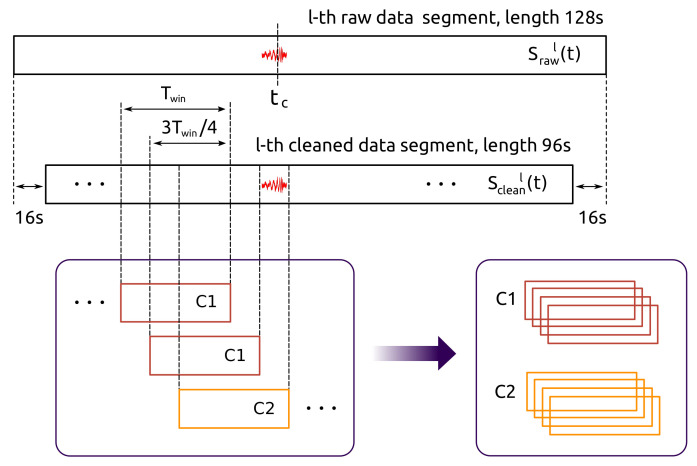
Strain samples generation by slidding windows. A CBC GW injection is located at coalescence time $t_{c}$ in l−th raw strain data segment of 128 s. After a cleaning, the segment is splitted in overlapped windows, individually identifying whether a window has or does not have the GW injection. Finally, among all these windows, and discarding C1 windows with SNR<10, a set of eight strain samples is selected, four with noise alone (C1) and four with noise plus the mentioned CBC GW injection (C2). This set of eight samples will be part of the input dataset of our CNNs. This procedure is applied to all raw strain data segments.

**Figure 4 sensors-21-03174-f004:**
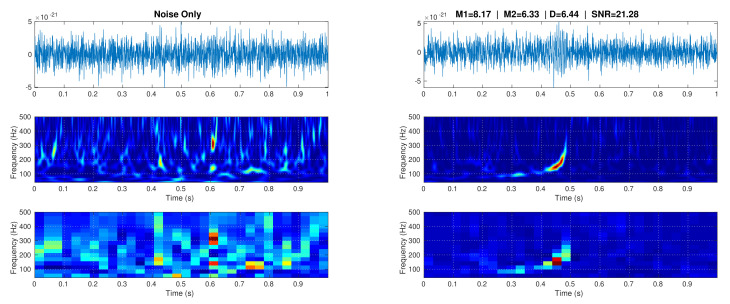
Visualization of two image samples of $T_{win}=1.0$ s, generated from a strain data segment of length 4096 s and GPS initial time 932335616, recorded at the L1 detector. In the left panel, a sample of noise alone (class 1) is shown, and in the right panel a sample of noise plus GW (class 2). In the class 2 sample, masses M1 and M2 are in solar masses units, distance *D* in Mpc, and SNR is the expected signal-to-noise ratio. Upper panel show strain data in the time domain, middle panels show time-frequency representations after apply a WT, and bottom panels show resized images by a bicubic interpolation.

**Figure 5 sensors-21-03174-f005:**
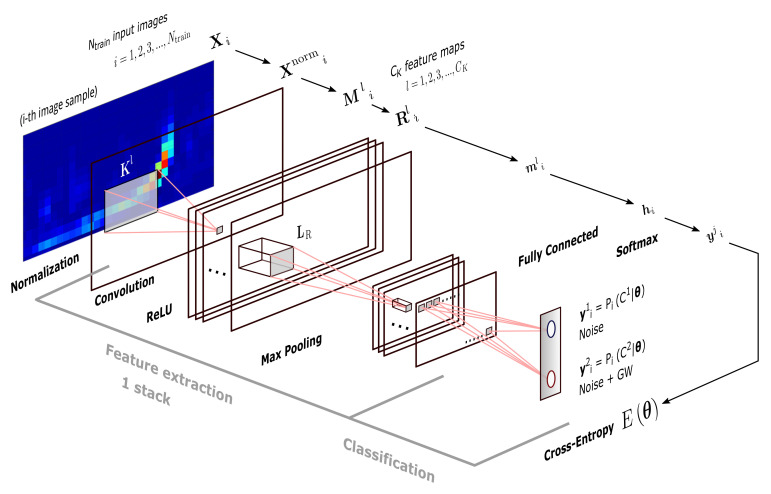
Single stack CNN architecture used as basis for this research. $N_{train}$ same size image samples are simultaneously inputted, but, for simplicity, we detail the procedure for a single image—besides, although this image is shown colorized, training images dataset occupies just one channel. A $X_{i}$ image feed the CNN, and a two-dimensional vector is outputted, giving us posterior probabilities of class 1 (noise alone) and class 2 (noise + GW), both being conditioned by model parameters included in vector θ. After that, cross-entropy $E\left(\theta \right)$ is computed. The notation for input(output) matrices and vectors is the same as that introduced in Section 2.5 to mathematically describe the several kinds of layers that are used in the CNNs.

**Figure 6 sensors-21-03174-f006:**
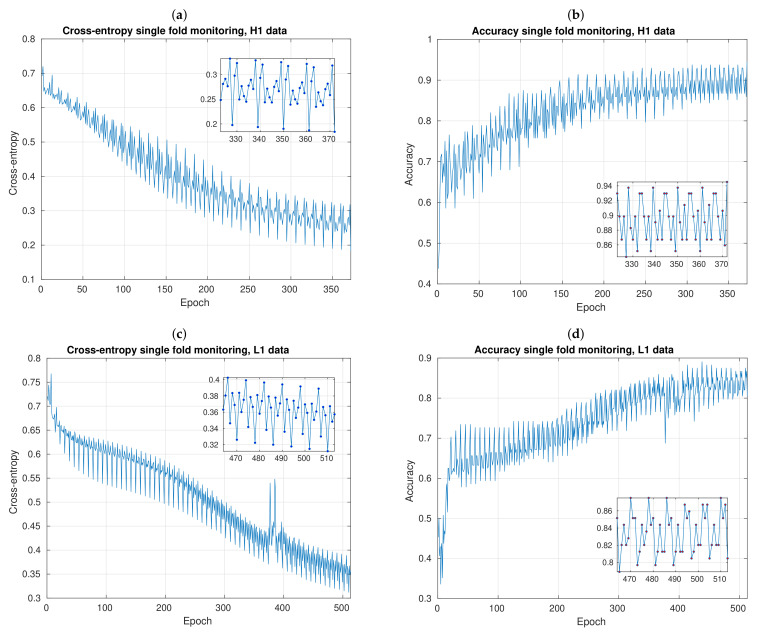
Evolution of cross-entropy (left panels: (**a**,**c**)) and accuracy (right panels: (**b**,**d**)), in the function of epochs of learning process, using data from H1 detector (upper panels) and L1 detector (lower panels). On trending, cross-entropy decreases and accuracy increases, even if there is a clear stochastic component due to the mini-batch SGD learning algorithm. Here we set length of sliding windows in $T_{win}=0.50$ s, and the CNN architecture with 2 stacks and 20 kernels. Some anomalous peaks appear when data from L1 detector is used, but this does not affect the general trends of mentioned metrics.

**Figure 7 sensors-21-03174-f007:**
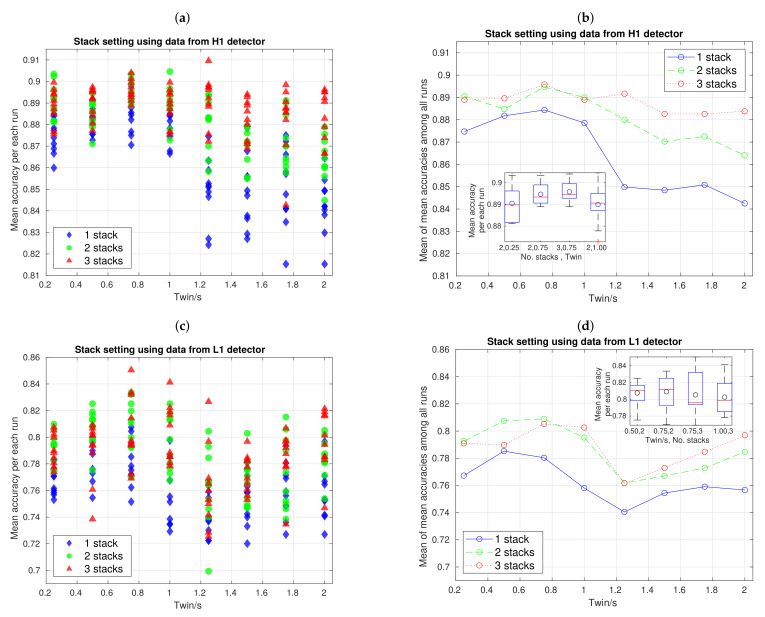
Hyperparameter adjustment to find best CNN architectures, i.e., number of stacks, and time resolution $T_{win}$. Left panels, (**a**,**c**), show all samples of mean accuracies, and right panels, (**b**,**d**), the mean of mean accuracies among all runs; all in function of $T_{win}$ for 1, 2, and 3 stacks in the CNN. Besides, small boxplots are included inside right plots to have more clear information about dispersion and skewness of distribution of mean accuracies that are more relevant and from where we chose our optimal settings. Based on this, we concluded that $T_{win}=0.75$ s, with 3 stacks (H1 data) and with 2 stacks (L1 data), are optimal settings. We used 20 convolutional kernels, repeating a 10-fold CV experiment 10 times.

**Figure 8 sensors-21-03174-f008:**
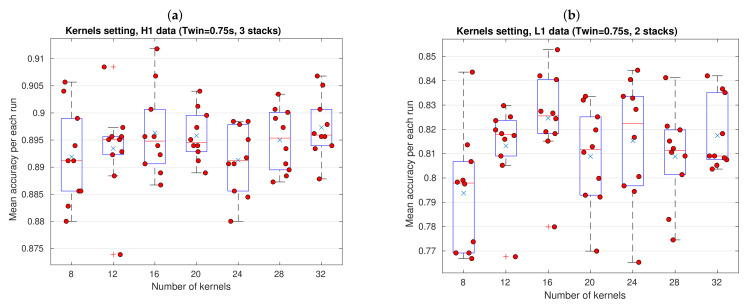
Adjustment to find the optimal number of kernels in CNNs for H1 data (**a**) and L1 data (**b**). Both panels show boxplots for the distributions of mean accuracies in function of the number of kernels, with values of $T_{win}$ and mount of stacks found in previous hyperparameter adjustments. Based on location of mean accuracy samples, dispersion, presence of outliers, and skewness of distributions, CNN architectures with 32 kernels and 16 kernels are the optimal choices to reach the highest mean accuracy values, when working with data from the H1 and L1 detectors, respectively.

**Figure 9 sensors-21-03174-f009:**
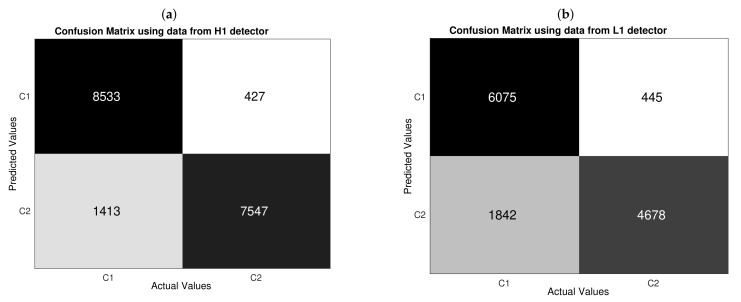
Confusion matrices computed from the testing for H1 data (**a**) and L1 data (**b**). C1 is the label for noise only and C2 the label for noise plus GW injection. Our CNN has 32 kernels and 3 stacks with H1 data, and 16 kernels and 2 stacks with L1 data. Time resolution is $T_{win}=0.75$ s. Working with H1 data, 8533/(8533+427)≈0.952 predicitions for C1 are correct and 1413/(8533+1413)≈0.142 predictions for C2 are incorrect and, working L1 data, 6075/(6075+445)≈0.932 predicitions for C1 are correct and 1842/(6075+1842)≈0.233 predictions for C2 are incorrect. These results show that our CNN very precisely classifies noise samples at the cost of reaching a not less number of false GW predictions.

**Figure 10 sensors-21-03174-f010:**
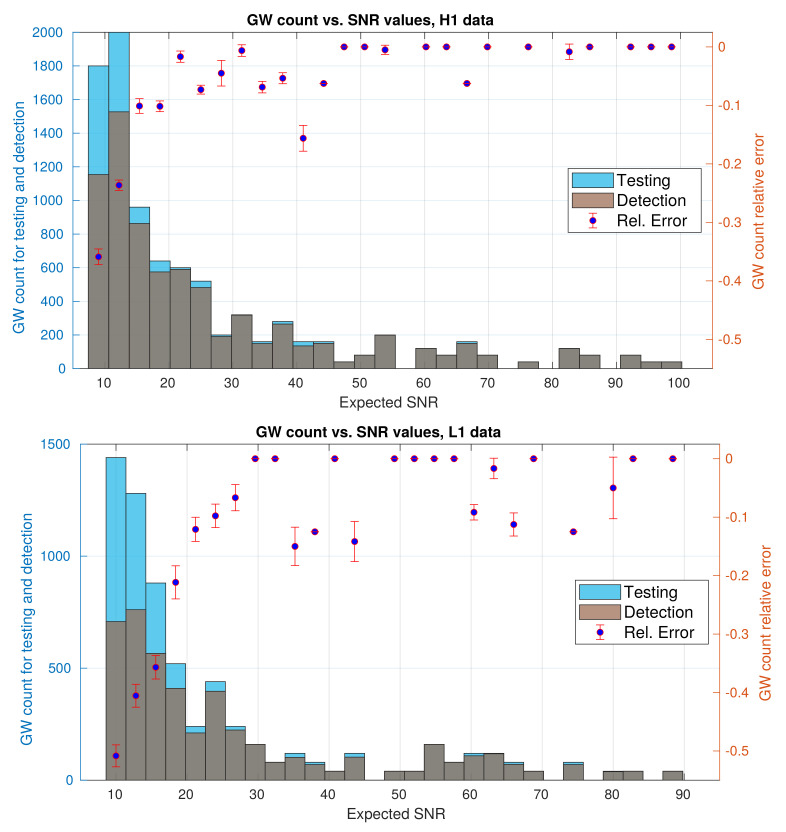
Histograms for counting GW samples present in test set and GW samples detected by the CNN. This count was made from all $100\;N_{test}$ predictions because we have 10×10=100 SGD learning-testing runs. Besides, the relative error between both histograms is shown as scatter points, with their respective standard deviations being computed from the 10 repetitions of the 10-fold CV routine. From plots, we have that our CNN detects more CBC GW events insofar as they have a SNR≥21.80 for H1 data (upper panel) and SNR≥26.80 for L1 data (bottom panel).

**Figure 11 sensors-21-03174-f011:**
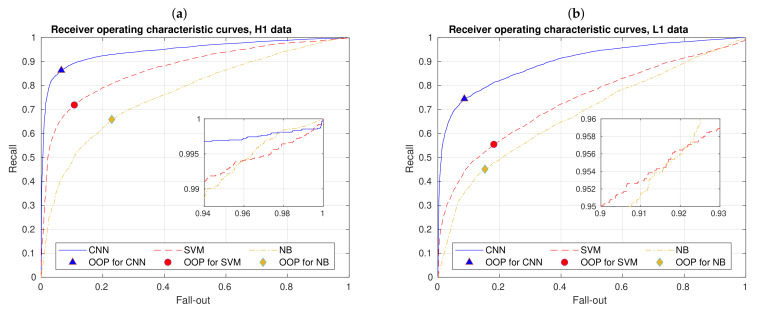
ROC curves for the CNNs, a SVM classifier with a linear kernel, and a Gaussian NB classifier. Data from H1 detector (**a**) and L1 detector (**b**) were used. For each ROC curve, its optimal operating point (OOP) is also shown. We set $T_{win}=0.75$ s and same CNN hyperparameter adjustments used in previous studies. The general trend, for almost all thresholds, is that our CNN has the best performance, followed by the SVM classifier, and finally by the NB classifier. Zoomed plots show small changes in performances near point (1,1), even though these do not affect the general trend.

**Figure 12 sensors-21-03174-f012:**
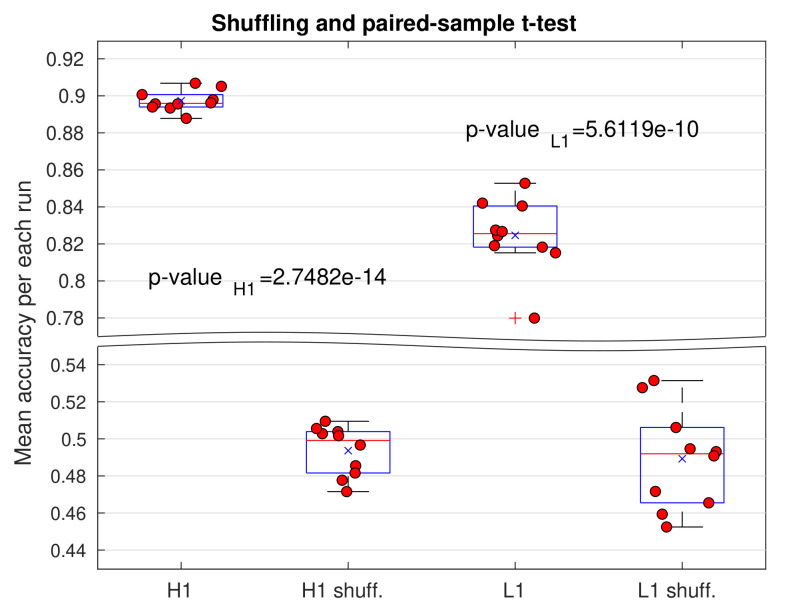
Shuffling and paired-sample t-test for elucidating if predictions of our CNN are statistically significant. Boxplots for the mean accuracies resulting of each experiment of the 10-fold CV, are shown. By shuffling, we mean that our training set is shuffled before each training, such that links between each sample and its known class is broken. Besides, the p-values of the mentioned paired-sample t-test are included, showing that predictions of the CNN are significantly different from a totally random process—with a significance level of 5%.

**Figure 13 sensors-21-03174-f013:**
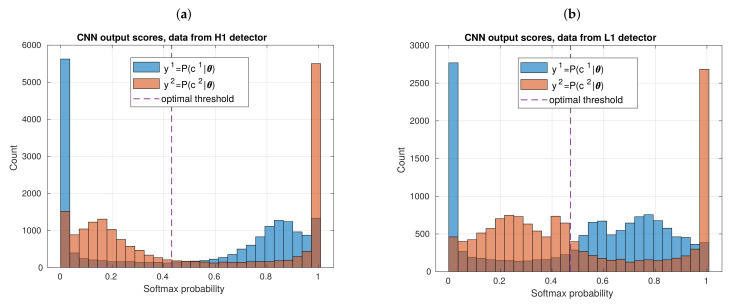
Histogram of CNN output probabilistic scores, $y^{1}$ and $y^{2}$, working with data from H1 detector (**a**) and from L1 detector (**b**). As it can be observed from the skewness towards edge bins, the CNNs are more optimistic predicting GW samples than predicting noise alone samples. Besides, with H1 data, there is less uncertainty than with L1 data, because, with L1 data, there is more population of occurrences in middle bins. Here, we counted all $100\;N_{test}$ predictions.

**Table 1 sensors-21-03174-t001:** The CNN architecture of 3 stacks used for this research. The number of kernels in convolutional layers is variable, and it took values $C_{K}\in \left\{8,12,16,20,24,28,32\right\}$. Part of this illustration is also valid for CNN architectures of 1 or 2 stacks (also implemented in this research), in which the image input layer is followed by the next three or six layers as feature extractor, respectively, and then by the fully connected layer until the output cros-entropy layer in the classification stage. All of these CNNs were implemented with the MATLAB Deep Learning Toolbox.

Layer	Activations per Image Sample	Learnables per Image Sample
**Image Input**	32×16×1	–
**Convolution** of size 5×4, $C_{K}$ kernels Strides: 1, Paddings: 0	$28\times 13\times C_{K}$	Weights: $5\times 4\times 1\times C_{K}$ Biases: $1\times 1\times C_{K}$
**ReLU**	$28\times 13\times C_{K}$	–
**Max Pooling** of size 2×2 Strides: 2, Paddings: 0	$14\times 6\times C_{K}$	–
**Convolution** of size 5×4, $C_{K}$ kernels Strides: 1, Paddings: 0	$10\times 3\times C_{K}$	Weights: $5\times 4\times C_{K}\times C_{K}$ Biases: $1\times 1\times C_{K}$
**ReLU**	$10\times 2\times C_{K}$	–
**Max Pooling** of size 2×2 Strides: 2, Paddings: 0	$5\times 1\times C_{K}$	–
**Convolution** of size 4×1×, $C_{K}$ kernels Strides: 1, Paddings: 0	$2\times 1\times C_{K}$	Weights: $4\times 1\times C_{K}\times C_{K}$ Biases: $1\times 1\times C_{K}$
**ReLU**	$2\times 1\times C_{K}$	–
**Max Pooling** of size 2×1 Strides: 1, Paddings: 0	$1\times 1\times C_{K}$	–
**Fully Connected**	1×1×2	Weights: $2\times C_{K}$ Biases: 2×1
**Softmax**	1×1×2	–
**Ouput Cross-Entropy**	–	–

**Table 2 sensors-21-03174-t002:** Confusion matrix for a binary classifier and its consequent standard performance metrics. In general, to have a complete view of the classifier, it is suitable to draw on, at least, accuracy, precision, recall, and fall-out. The F1 score and G mean1 are useful metrics for imbalanced multilabel classifications, and they also give important moderation features that help in model evaluation. Each metric has its probabilistic interpretation.

	Metric	Definition	What Does It Measure?
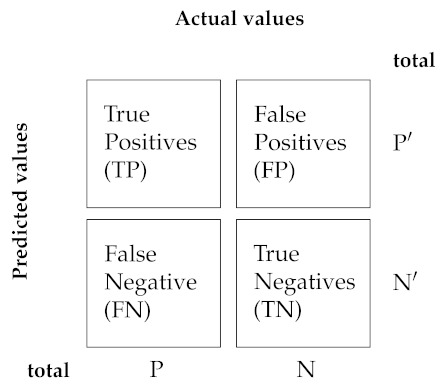			
	Accuracy	$\frac{TP+TN}{P+N}$	How often a correct classification is made
	Precision	$TP/P^{\prime }$	How many selected examples are truly relevant
	Recall	TP/P	How many truly relevant examples are selected
	Fall-out	FP/N	How many no relevant examples are selected
	F1 score	$2\frac{Precision\times Recall}{Precision+Recall}$	Harmonic mean of precision and recall.
	G mean1	$\sqrt{Recall\times Fall-out}$	Geometric mean of recall and fall-out.

**Table 3 sensors-21-03174-t003:** Standard metrics computed from values of confusion matrices shown in Figure 9. We set our CNN with 32 kernels and 3 stacks with H1 data, and with 16 kernels and 2 stacks with L1 data. The time resolution is $T_{win}=0.75$ s. Except for the G mean1 reaching moderate results that could help to avoid over pessimistic (or optimistic) results, accuracy, precision, recall, fall-out, and F1 score showed that the CNN has a better performance working with H1 data.

Standard Metrics with H1 Data				
Metric	Mean	Min	Max	SD
Accuracy	0.897	0.888	0.907	0.00564
Precision	0.952	0.941	0.968	0.00852
Recall	0.858	0.844	0.873	0.00894
Fall-out	0.0534	0.0374	0.0647	0.00880
F1 score	0.903	0.894	0.912	0.00509
G mean1	0.213	0.179	0.237	0.0185
**Standard Metrics with L1 Data**				
Metric	Mean	Min	Max	SD
Accuracy	0.825	0.780	0.853	0.0198
Precision	0.932	0.905	0.956	0.0161
Recall	0.768	0.724	0.793	0.0197
Fall-out	0.0869	0.0560	0.127	0.0208
F1 score	0.842	0.804	0.866	0.0168
G mean1	0.256	0.211	0.303	0.0282

**Table 4 sensors-21-03174-t004:** Additional metrics that were computed from ROC curves in Figure 11. From the area under the ROC curve (AUC), we have that, for both datasets, the CNN has the best performance (the highest AUC value), followed by the SVM with a middle performance, and finally by the NB classifier with the worst performance. Notice that, for the CNN, better performance does not mean an optimal threshold that is closer to the default 0.5 value.

Data	Model	Optimal Operating Point	Optimal Threshold	Optimal ${\mathrm{cost}}_{\exp}$	AUC
H1	CNN	$\left(0.0654,0.863\right)$	0.430	0.0505	0.946
	SVM	$\left(0.108,0.719\right)$	0.228	0.0972	0.872
	NB	$\left(0.229,0.659\right)$	0.379	0.143	0.768
L1	CNN	$\left(0.0860,0.744\right)$	0.472	0.0854	0.897
	SVM	$\left(0.182,0.555\right)$	0.418	0.157	0.736
	NB	$\left(0.153,0.451\right)$	0.920	0.175	0.681

## Data Availability

MATLAB source code used for this study, https://doi.org/10.5281/zenodo.4722120, doi: 10.5281/zenodo.4722120.

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
