# Peer review of "Deep Learning for Gravitational-Wave Data Analysis: A Resampling White-Box Approach"

_sensors, 2021, doi:10.3390/s21093174_

Round 1

Reviewer 1 Report

This is an interesting paper concerning gravitational wave data analysis. The paper is well written, well referenced and seems mathematically correct. Therefore, it deserves to be shared with the Scientific Community. Thus, publication is recommended.

Author Response

We thank to the reviewer for his/her report and approval of the manuscript.

Reviewer 2 Report

Review of " Deep learning for gravitational-wave data analysis: A resampling white-box approach" by Manuel D. et al.

General Comment:

The prime objective of this manuscript is to provide a deep learning algorithm to separate the gravitational wave signals from the noise in LIGO data. Key results from the comprehensive analysis indicate that the softmax layer affects the accuracy of CNNs algorithms. In my opinion, the authors’ work has two main contributions. It’s more realistic to work only with data of non-Gaussian noise when testing the algorithms. On the other hand, by comparing with NB and SVM models, the advantages of CNNs in terms of accuracy are better demonstrated.

With some effort on the part of the authors to improve the presentation, I believe it deserves publication. I attach comments below that I hope will be helpful to the authors and editors.

Major Comments:

  1. The introduction section is too much about the algorithm and gives too much details in this paper. I think it is better to generalize the language a bit, and the specific algorithmic advantages of this paper can be introduced subsequently.
  2. There is no uniformity in italics and bold for article formulas and symbols, and I suggest using italics uniformly.

Specific Comments:

  1. P.2 Line 43: ‘powerfull’. Misspelling of words.
  2. P.6 Line 241: Discarding missing data directly does not seem to be a very rigorous approach, indicating that the dataset is not fully utilized. I suggest to do some pre-processing as much as possible.
  3. P.7 Line 263: The number ‘2’ on the word function is confusing.
  4. P.16 Line 500: “But even more, following…” The reason for choosing k = 10 to get a better result is not fully articulated. The use of the but transitive statement is confusing.
  5. P.16 Line 513: The use of ‘anyway’ is a bit colloquial, so I suggest replacing it with in any case or in short.
  6. P.19 Figure 6: The detail figures in Figure 6 can be placed more aesthetically pleasing in the lower left corner. I suggest adding alphabetical numbers to the top left corner of the four figures to make the descriptions easier.
  7. P.21 Figure 7: The same recommendation is made as in Figure 6. It’s better to add alphabetical numbers.

Author Response

We thank the Reviewer for their constructive comments and suggestions, that have helped us to improve our manuscript. A summary of the specific responses is outlined below.

Major comments:

1) We are totally agree with this comment. Then, we moved motivations, advantages, and technical details about our resampling white-approach to a new subsection in the paper (2.4 "Resampling and white-box approach"), from Line 349 to Line 405. A sentence was added from Line 111 to Line 113 to make a connection with the new subsection 2.4. Moreover, the last paragraph of the Introduction was changed and, most of it was moved to the subsection 2.2., from Line 173 to Line 210. Finally,  a couple of sentences were added to the end of the introduction (line 134 to 137, ) to connect ideas with subsection 2.2.

2) We accept the suggestion about italics style. Now, all formulas are in italic, including subscripts (that initially were in normal text), and even units if they follow a number (as Hz, Mpc, etc.). On the other hand, we use bold in equations following an standard notation that is usual in mathematical literature: all quantities of 2 or more dimensions (as vector and matrices) are written in bold, and scalars in normal style.

Specific comments:

1) Corrected.

2) We thank very much this criticism, because it helped us to write better the paragraph; and even, allowed us to found a mistake. It is true that discarding missing data directly it is not a rigorous approach in many cases. However, in our case, we have strong reasons to perform this; and it is because the NaN entries span time ranges than are greater than resolutions than we apply for building our strain samples that are inputted by the CNN. Feature engineering procedures for dealing with missing values (as imputation and extrapolation) are not suitable in this scenario. On the other hand, the reason to discard blind injections (included in log files given by LIGO, different from strain data files), is because we need information about injections (particularly, expected SNR values) to do the post-processing included in Fig. 10 -in any case, to address the problem of parameter estimation is beyond the scope of our work. From Line 237 to 245 we modified the text, in order to clearly explain the above reasons for discarding missing data, indicating the time ranges spanned by NaN values and time resolutions (T_{win}). In addition, from Line 224 to 228 we included an additional change to correct the mistake that we found. The point here is that we downloaded 722 data segments for H1, and 652 data segments for L1 (all of time length 4060s as it is mentioned in Line 169), and only after the first step of the cleaning (extraction of segments of length 128s with hardware injections and rejection of segments that included NaN values), these data are decreased to 501 segments for H1, and 402 segments for L1. These last numbers was that we wrongly mentioned in the previous version of the manuscript, as the number of files that we downloaded.

3) We moved the number 2 (now 1) after the period, Line 255.

4) We changed the sentence "But even more" by "Moreover", now located in Line 550. Besides, the choice of k=10 in k-fold CV is motivated by references [63-65], in Line 549. This empirical number is not a hyperparameter (to be tuned) either can be obtained by mathematical proofs; then we decided to motivate our choice in previous works of ML community.

5) We changed the word "anyway" along all the paper by "in any case" in some places, and by "in short" in other places, as suggested.

6) and 7) Unfortunately, due to LaTeX automatic pagination, we was unable to locate Figure in the lower left corner. However, we were able to add alphabetic letter identification to each plot in the left top corner, as suggested. Body text was slightly changed to mention these letters, Lines: 640, 641, 656-658, 696, 699, 705, 711-712, 719, 731, 742-743, 745. We also included changes in captions of Figures 6 and 7.

Reviewer 3 Report

The author present a novel "white-box" resampling approach with CNNs, novel for the use of gravitational-wave detection. While it is not competitive with some of the other existing methods, this paper serves as an exhaustive and pedagogical text that should be useful for anyone entering this field. Towards this end, I might encourage the authors to strongly considering releasing the MATLAB scripts they used, as it would be useful for others attempting to enter this field. After some comments below (especially about the emphasis on the benefits of using S6 data), I think it should be published.   Major comments: I think more discussion of the choice of S6 data should be made. Quantitatively and qualitatively the data has changed, and arguing that this choice makes for the most adverse conditions seems like a weak one, relative to choosing publicly available O2 data. There is also a reason we moved away from hardware injections: We have long known that our hardware injections produce statistically consistent values with software injections, but lack for realistic populations (due to limited time that we can modify otherwise usable data with them). I do not think this is somehow a strength for the paper that software injections are not being used, as seems to be implied in the abstract; the population distributions we know about are drastically different from S6.   Minor Comments: 27: KAGRA should be briefly mentioned. 42: Should probably call out gstlal and MBTA if going to call one. 107: Maybe cite the rates papers here? 120: the authors should discuss the use of Morlet here, compared to some of the other common choices in spectrogram based analyses in the LVK. 285: Can the authors talk about the assumption behind what "injection present" means? Some fractions of segments presumably have portions of the time-series buried in noise such that they could never be detected (SNR << 1). Is it useful to label these as "injection"? Figure 10: Can the authors discuss more why the behavior is not monotonic? Is it just due to the choice of binning?   Typo: 30: stirring? 37: asumption  644: hiperparameter  770: profient 

Author Response

We appreciate constructive criticisms from the Reviewer, which allowed us to importantly improve our manuscript. A summary of the specific responses is outlined below. Thank again.

Source code:

Following the suggestion of the Reviewer, we decided to release the MATLAB code. Now it is available on GitHub: https://github.com/ManuelDMorales/dl_gwcbc. In the paper, we included this link as reference in Line 129.

Major comments:

We expanded the discussion about our choice of S6 data, and moved it from the introduction to the subsection 2.2., from Line 173 to Line 210. The most important reason why we used data from S6 run, is because it has a much greater amount of hardware injections of CBC GW signals than in later public data (as O1 and O2); and the most important reason for working solely with hardware injections, lies on the fact that it is important to set beforehand a constraint to avoid having control on distribution of the initial dataset based on ad hoc choices, which in turn could influence performance results. We agree with the Reviewer about the lack for realistic populations when hardware injections are involved, but the point is that our approach is based on the constraint to generate the population, not on the population itself. This advantage cannot achieved with software injections, which even random, these are still handleable and, therefore, evaluation results can be influenced. As now is mentioned in the manuscript, problem of bias and fairness regarding datasets and algorithms is an open and highly debate issue in ML community, which even has deep ethical issues when we are talking about applications of ML/DL in social domains. Therefore, it would be important that in GW data analysis community we begin to be aware about the importance of set our evaluation procedures as transparent as possible. In any case, even if above advantage is not considered, our approach still is scientifically relevant; because it lays the groundwork for future calibration procedures with CNN detection algorithms inputting hardware injections.

Minor Comments:

1) We now mention KAGRA in Line 29.

2) We now mention GstLAL and MBTA pipelines in Line 43, with references.

3) In Line 359 we included two recent references of LIGO about BBH populations from O1 and O2 scientific runs.

4) We have added a brief discussion, from Line 114 to Line 120, to clarify why our choice of Morlet wavelet as mother wavelet. The main idea is that we want to have all time-frequency at once, which removes beforehand other wavelet options that have been used for multiresolution analyses.

5) In our pipeline, after we label segments of class 1 data (injection present), only those samples of SNR>10 are considered for next steps in the preprocessing. We included this information in Line 280, and in caption of Fig. 3. Sorry for this omission.

6) In Fig. 10, bin-by-bin discrepancies are explained, mainly, because our predictions are stochastic; even if binning could also affect a bit. We included an additional discussion from Line 905 to Line 910.

7) This word is incorrect. Then, we changed "stirring" by "minimizing". Line 32.

8) Word "asumption" was corrected, now is "assumption". Line 38.

9) Word "hiperparameter" was changed by "hyperparameter". Line 694.

10) Word "profient" was changed by "proficient". Line 821.